# StyleGuard: Preventing Text-to-Image-Model-based Style Mimicry Attacks by Style Perturbations

**Yanjie Li, Wenxuan Zhang, Xinqi Lyu, Yihao Liu, Bin Xiao** [*]
Department of Computing, Hong Kong Polytechnic University
{yanjie.li, leo02.zhang, xinqi.lyu, yihao5.liu} @connect.polyu.hk
b.xiao@polyu.edu.hk

## Abstract

Recently, text-to-image diffusion models have been widely used for style mimicry and personalized customization through methods such as DreamBooth and Textual Inversion. This has raised concerns about intellectual property protection and the generation of deceptive content. Recent studies, such as Glaze and Anti-DreamBooth, have proposed using adversarial noise to protect images from these attacks. However, recent purification-based methods, such as DiffPure and Noise Upscaling, have successfully attacked these latest defenses, showing the vulnerabilities of these methods. Moreover, present methods show limited transferability across models, making them less effective against unknown text-to-image models. To address these issues, we propose a novel anti-mimicry method, StyleGuard. We propose a novel style loss that optimizes the style-related features in the latent space to make it deviate from the original image, which improves model-agnostic transferability. Additionally, to enhance the perturbation's ability to bypass diffusion-based purification, we designed a novel upscale loss that involves ensemble purifiers and upscalers during training. Extensive experiments on the WikiArt and CelebA datasets demonstrate that StyleGuard outperforms existing methods in robustness against various transformations and purifications, effectively countering style mimicry in various models. Moreover, StyleGuard is effective on different style mimicry methods, including DreamBooth and Textual Inversion. The code is available at `https://github.com/PolyLiYJ/StyleGuard`.

## 1 Introduction

Diffusion models have demonstrated remarkable effectiveness in various applications, such as image generation [13], image editing [22], and text-to-image synthesis [43, 25]. The emergence of diffusion models has significantly transformed the art industry [28]. These models allow users to create detailed artwork from simple text prompts, a task that once required extensive time and skill from professional artists. However, these technologies have also raised concerns about copyrights and ethics. For example, Dreambooth [36] allows anyone to fine-tune an SD model to imitate an artist's style with just a few paintings and generate high-quality artwork. This seriously damages the intellectual property rights of artists.

To tackle the challenges associated with unauthorized image usage in text-to-image generation, recent perturbation-based approaches have emerged. These methods are designed to subtly modify user images, rendering them "unlearnable" for malicious applications and disrupting the functionality of targeted diffusion models. For example, Anti-Dreambooth [40] shows some effectiveness by alternately training diffusion models and executing PGD attacks, but it is fragile against simple data transformations. MetaCloak [29] builds on Anti-Dreambooth by incorporating simple transformations

---

[*]Corresponding Author

39th Conference on Neural Information Processing Systems (NeurIPS 2025).

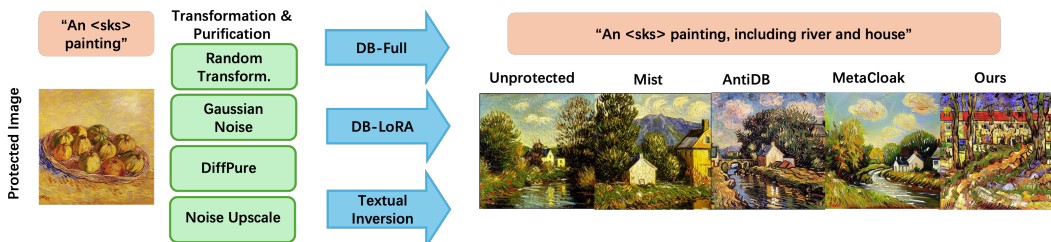

Figure 1: A comparison of the defensive performance of different methods in the presence of the purification transformations. Previous methods, including Mist, AntiDreamBooth, and MetaCloak, fail to defend against DiffPure and Noise Upscale, while our proposed method successfully resists the style mimicry attack under various transformations for different customization methods.

into the attack pipeline. SimAC [41] improves MetaCloak by selecting time steps with the highest gradients to improve the stability of the training.

Although these methods have demonstrated effectiveness against style mimicry, recent purification-based methods pose significant challenges to these protections. For example, DiffPure [34] and Noise Upscaling [19] utilize LDM-based purifiers and upscalers to purify noise and successfully bypass the defenses of these methods, as illustrated in Figure 1. These novel attacks highlight the vulnerabilities of current anti-customization techniques. Moreover, in the real world, the finetuning model may differ from the model used in the training phase, making cross-model transferability an important issue. In light of these limitations, there is a need for a more efficient method to generate protective noise that *is resilient to diffusion-based purifications* and *has strong cross-model transferability*.

To solve these problems, we propose StyleGuard, a more effective method based on style perturbations to protect artists from unauthorized text-to-image diffusion-based style mimicry. To generate transferable perturbations, inspired by style transfer [17] and adversarial attacks in feature space [44], we disrupt style-related features in latent space by adding subtle noise to the fine-tuning images that is nearly imperceptible to the human eye. Our method prevents text-to-image models from accurately extracting the style features of fine-tuning images, leading to false style correlations and hindering attackers from mimicking the original image's style. Moreover, compared to Mist [26] that uses $L_2$ distance in the feature space, which is prone to causing local disturbances and sensitive to image variations, style perturbations alter global features. As a result, our method is more robust to transformations such as image cropping, rescaling, and Gaussian noise. Additionally, bypassing diffusion-based purifications presents significant challenges, particularly because directly incorporating purifiers or upscalers into the optimization process can lead to memory overflow, as most purifiers and upscalers are based on diffusion models [34, 24, 7]. To address this issue, we propose a novel upscale loss function that maximizes the loss of denoise error on ensemble purifiers and upscalers through a meta-learning approach. On the WikiArt and ClebA-HQ datasets, we show that StyleGuard significantly outperforms existing approaches over various transformations and purification methods and has higher cross-model transferability. Our main contributions are summarized as follows.

- We propose StyleGuard, a robust method designed to effectively protect artists from DreamBooth-based style mimicry. StyleGuard accounts for various preprocessing techniques that attackers may employ, enhancing its practical effectiveness.

- To improve the model-agnostic transferability, we introduce a novel style loss, which aligns the style characteristics of the protected image more closely with those of the target image, allowing the fine-tuned model to establish an incorrect style connection.

- To bypass purification transformations, we propose a novel upscale loss function to maximize the denoise-error loss on the ensemble purifiers and upscalers. Experimental results show that our approach exhibits strong robustness against the latest purification methods, including DiffPure and Noise Upscaling, even on unseen purifiers.

- Experiments on the WikiArt and CelebA datasets demonstrate that StyleGuard offers enhanced protection against style mimicry and identity customization.

- Compared to previous defense methods, our approach shows improved efficacy and practical effectiveness by considering various preprocessing techniques and model-agnostic scenarios.

## 2 Background

Diffusion models have emerged as a powerful tool for image generation. Recent works have demonstrated their effectiveness in various domains, such as inpainting, super-resolution, and style transfer [9, 30, 16, 35, 31]. The flexibility of diffusion enables the generation of high-quality images from textual descriptions.

### 2.1 Style Mimicry and Copyright Concerns

Unauthorized style mimicry has become a major concern in the AI art community, where malicious actors exploit AI models to replicate an artist's unique style without consent [4, 6, 32]. Advanced mimicry attacks involve fine-tuning generic text-to-image models on a small collection of an artist's works, often as few as 20 pieces by models such as DreamBooth [10, 36]. DreamBooth identifies key stylistic features and associates them with a specific token in the fine-tuned model, enabling highly accurate style replication. Such techniques have led to widespread incidents of unauthorized mimicry [4, 6, 32], for example, CivitAI [8] built a large online website where people share their finetuned stable diffusion models. The potential for unauthorized style mimicry threatens the livelihoods of artists, leading to discussions about intellectual property rights in the digital age.

### 2.2 Protection Against Style Mimicry and Personalization

To address the unauthorized style mimicry issue, perturbation-based methods have been developed, which add subtle image perturbations to the unprotected images to disrupt generative models. For example, PhotoGuard [37] aligns protected images' latent features with black-and-white images. Glaze [38] minimizes the feature distance between perturbed images and target images while maintaining perceptual similarity. AdvDM [27] reduces the likelihood of perturbed images under pre-trained diffusion models by disrupting the denoising process. Its enhanced version, Mist [26], utilizes black-and-white periodic images as targets and incorporates semantic loss and textual loss to improve protective strength. However, Mist directly minimizes the L2 distance between the original and target latent features, which causes noticeable and unnatural textures that degrade the original image's quality. Anti-DreamBooth [40] proposes a novel scheme to defend the personalization attack that alternately updates the diffusion model and protected images. MetaCloak [29] builds upon Anti-DreamBooth by incorporating simple transformations, such as Gaussian blur and cropping, into the attack pipeline to improve robustness against these transformations. SimAC [41] further extends the work done by Anti-DreamBooth by selecting timesteps with maximum gradients to stabilize the training process. Some other methods [15, 46, 45, 42] protect copyrights by embedding watermarks into images, subtly incorporating the author's information. However, this kind of approach introduces additional verification processes. While these methods exhibit robustness to simple transformations, recent work has introduced purification-based methods, including DiffPure [34] and Noise Upscaling [19], which first add noise to images and then employ an LDM as purifier or upscaler to remove noise. These approaches have demonstrated impressive results in removing protective noise, rendering many recent protective methods ineffective. Therefore, there is an urgent need to develop more robust methods to defend against such attacks.

## 3 Preliminary

### 3.1 Style Mimicry by Dreambooth

DreamBooth [36] introduces a novel approach for personalizing text-to-image diffusion models by enabling them to generate high-fidelity images of specific subjects based on a few reference images. The method fine-tunes a pre-trained text-to-image model to bind a unique identifier (e.g., "[V]") to the subject, allowing the model to synthesize the subject in diverse contexts while preserving its key visual features. This is achieved through a fine-tuning process: the latent diffusion model (LDM) is fine-tuned using input images paired with text prompts containing the unique identifier and the subject's class name (e.g., "An [V] painting"), while a class-specific prior preservation loss ensures the model retains its semantic understanding of the broader class (e.g., "A painting"). The DreamBooth loss function is defined as

$$\mathcal{L}_{gen}(\theta, x_0) = \mathbb{E}_{x_0, t+1|\epsilon} \|\epsilon - \epsilon_0 (x_{t+1}, t, c)\|_2^2 + \lambda \|\epsilon_0 (x_{t+1}, t', c_{pr})\|_2^2 \quad (7)$$

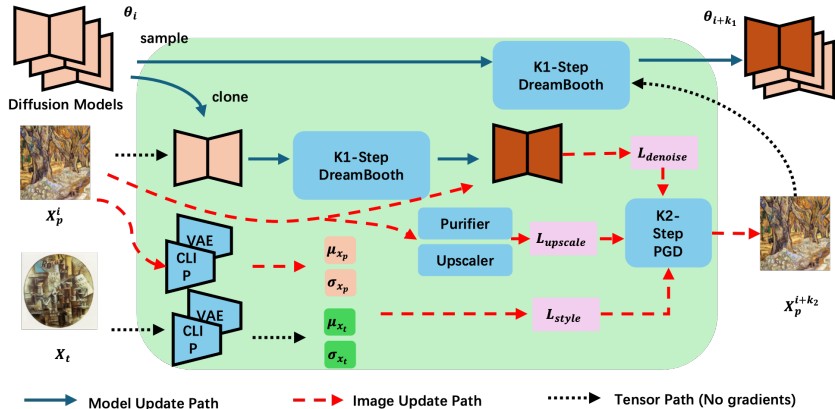

Figure 2: The pipeline of StyleGuard. We alternatively update the diffusion model and the protected images. Ensemble image encoders and purifiers are included to compute the style loss and upscale loss to improve cross-model transferability and the robustness to purifications.

By leveraging the model's semantic prior and the text-guided denoising loss function, DreamBooth enables tasks like subject recontextualization, text-guided view synthesis, and artistic rendering, overcoming limitations of existing text-to-image models in reconstructing and modifying specific subjects.

## 4 Methodology

The pipeline of StyleGuard is shown in Figure 2. We will first define the problem and then introduce the loss functions. Finally, we will introduce the StyleGuard Algorithm.

### 4.1 Problem Statement

We frame the problem as follows: a user aims to safeguard a set of clean and unprotected images $X_c = \{x_c^i\}_{i=1}^n$ from being exploited by unauthorized model trainers for generating style-mimicking images. To accomplish this, the user applies a small perturbation to $X_c$, resulting in a modified set of protected images $X_p = \{x_p^i\}_{i=1}^n$, which can be safely released on the Internet. Adversary will then gather and utilize $X_p$ to fine-tune a text-to-image generator $\theta$ following the DreamBooth algorithm or any other methods. We assume that the model trainer has some awareness of the protection and tries to destroy the protection effectiveness through different pre-processing methods, such as random transformations and purifications to the training image set $X_p$.

The goal of the user is to create a protected and robust image set $X_p$ that diminishes the personalized generation capabilities. This objective can be expressed as a bilevel optimization problem:

$$X_p^* \in \arg \max_{X_p, \theta^*} L_{\text{dis}}(X_{gen}; X_c) \tag{1}$$

$$\text{where} \quad \theta^* \in \arg \min_{\theta} \left\{ L_{\text{gen}}(\mathbb{T}(X_p); c, \theta) \right\}. \tag{2}$$

In these equations, $c$ denotes the class-wise conditional vector. $X_{gen} = M_{\theta^*, X_p}(c)$ is the generated images of the fine-tuned LDM model $M_{\theta^*, X_p}$. $L_{\text{dis}}$ represents a perception-aligned distance function used to evaluate the style discrepancy between the generated images $X_{gen}$ and the clean reference images $X_c$. The $L_{\text{gen}}$ is the finetuning loss function. We hypothesize the adversary tries to destroy the protection of $X_p$ by applying some kinds of preprocessing methods (denoted by $\mathbb{T}$).

### 4.2 Disturbing Style-related Features and Generating Transferable Perturbations

Previous work has found that the mean and variance of the feature space encapsulate the style information [39, 44, 18, 20]. We consider the problem defined in Eq. 4.1 as maximizing the style-related feature distance between the reference images and the perturbed images and making it closer to the target images. Therefore, we propose a novel style loss function, which is defined as

$$L_{\text{style}} = \mathbb{E}_{f \sim F} \mathbb{E}_{x_p \sim X_p, x_t \sim X_t} \left( \left\| \mu_{x_p} - \mu_{x_t} \right\|_2^2 + \left\| \sigma_{x_p} - \sigma_{x_t} \right\|_2^2 - \left\| \mu_{x_p} - \mu_{x_c} \right\|_2^2 - \left\| \sigma_{x_p} - \sigma_{x_c} \right\|_2^2 \right), \tag{3}$$

where $X_t$ is a set of target images that has a different style from $X_c$. The $\mu$ and $\sigma$ are the mean and variance of the latent features of these images encoded by the LDM's VAE or CLIP encoder $f$. To improve the transferability to unknown models, we compute the style loss over a set of different substitute encoders. The target image is selected to have a distinct style from the original images, such as from different art movements (e.g., realistic and abstract). By disturbing the style-related features, StyleGuard can make it difficult for DreamBooth or Textual Inversion to establish a correct connection between the style features and the unique identifier.

### 4.3 Why Previous Defenses are Ineffective to Noise Upscaling and DiffPure

Previous defenses, such as MetaCloak [29] and SimAC [41], involve transformations like random cropping or Gaussian blur in the perturbation generation process to enhance robustness against preprocessing methods. However, these approaches fail to counter attacks like DiffPure [34] and Noise Upscaling [19], which use diffusion models as noise purifiers. Directly incorporating Noise Upscaling or DiffPure into the optimization process can result in an excessively large computation graph. To address this challenge, we first apply small noises to $x_p$ according to the DiffPure and Noise Upscale settings and then maximize the denoising error loss across a set of substitute DiffPure and upscaling (super-resolution) models. The upscale loss function is defined as

$$L_{\text{upscale}} = -\mathbb{E}_{\theta_{\mathbb{T}} \sim \Theta_{\mathbb{T}}} \mathbb{E}_{x'_{p,0}, t, c, \epsilon \sim \mathcal{N}(0,1)} \left\| \epsilon - \epsilon_{\theta_{\mathbb{T}}}(x'_{p,t+1}, t, c) \right\|_2^2, \text{ where } x'_p = x_p + \delta N(0,1), \tag{4}$$

where $\Theta_T$ is the parameter of purification and upscaling models. Upscale loss function aims to cause purifiers and upscalers to amplify protective perturbations instead of reducing it during the diffusion process. The upscale loss is computed at a randomly-chosen timestep in the denoising sequence. Still, this method is effective in breaking popular purifiers and upscalers (Sec.5.2)

### 4.4 The Algorithm of StyleGuard

A straightforward approach to tackle the bilevel problem in Section 4.1 is to unroll all training steps and optimize the protected examples $X_p$ through backpropagation. However, this would cause a very large computation graph that would exceed the capacity of most current machines. To overcome this challenge, inspired by Anti-DreamBooth [40], we use an approximate method to optimize the $X_p$ and $\theta$ alternately. Specifically, in the $t$-th iteration, when the current model weights $\theta_t$ and the protected image set $X_p^t$ are available (with $\theta_0$ initialized from a pretrained diffusion model and $X_p^0 := X_c$), we create a copy of the current model weights, denoted as $\theta'_{t,0} \leftarrow \theta_t$, for noise crafting. We then optimize the UNet of LDM for $K_1$ steps using DreamBooth loss:

$$\theta'_{t,j+1} = \theta'_{t,j} - \beta \nabla_{\theta'_{t,j}} L_{\text{gen}}(X_p^t; \theta'_{t,j}), \tag{5}$$

where $j \in \{0, 1, \ldots, K-1\}$ and $\beta > 0$ is the step size. This unrolling process enables us to "look ahead" during training and assess how current perturbations will influence the fine-tuned LDM.

Subsequently, we utilize the updated UNet model $\theta'_{t,K}$ to optimize the upper-level problem, specifically updating the protected images $X_p$ by PGD. However, it is difficult to update the training images $X_p$ of LDM by $L_{dis}$ directly because this needs to unroll the fine-tuning process. To make the gradient computable, we maximize the denoising-error loss $L_{\text{denoise}}$ instead. Moreover, in real-world scenarios, the pretrained text-to-image generator used by unauthorized model trainers is often unknown. To enhance the transferability of the perturbed images to unknown models, we alternatively maximize the denoising error across a group of LDMs. The denoise loss is defined as

$$L_{\text{denoise}} = -\mathbb{E}_{\theta \sim \Theta} \mathbb{E}_{x_{p,0}, t, c, \epsilon \sim \mathcal{N}(0,1)} \left\| \epsilon - \epsilon_\theta(x_{p,t+1}, t, c) \right\|_2^2 \tag{6}$$

The total loss function is defined as:

$$L_{\text{styleguard}} = L_{\text{denoise}} + \eta L_{\text{upscale}} + \lambda L_{\text{style}}, \tag{7}$$

where $\eta$ and $\lambda$ are hyperpameters. We set $\eta$=1 and $\lambda$=10 in our experiments. Moreover, to improve the robustness to other transformations like crop and resize, we involve random transformations $\mathbb{T}'$ in the optimization process based on the expectation over transformation (EoT) [29, 5]. Then we update the perturbed image $X_p$ using PGD with the attack budget $B_\infty$ for $K_2$ steps through

$$X_p^{t+1} = \mathbb{E}_{g \sim \mathbb{T}'} \left[ \Pi_{B_\infty} \left( X_p^t - \alpha \operatorname{sign} \left( \nabla_{X_p^t} L_{\text{styleguard}} \right) \right) \right], \tag{8}$$

This expectation is estimated by Monte Carlo sampling with $J$ samples ($J = 1$ in our setup). After obtaining the updated protected images $X_p^t + \delta^t$, the surrogate model $\theta_t$ is trained on the perturbed images for additional $K_1$ SGD steps:

$$\theta_{t+1} = \theta_t - \beta \nabla_{\theta_t} L_{\text{gen}}(X_p^t + \delta^t; \theta_t). \tag{9}$$

The procedures outlined above will repeat $N$ times, resulting in the final protected images $X_p^*$. The final algorithm is shown in Algorithm 1.

---

**Algorithm 1** The Algorithm of StyleGuard

---

1: **Input:** Initial substitute LDM model set $\Theta_0$, initial protected images $X_p^0 = X_c$, pretrained image encoders $F$, number of iterations $N$, target images $X_t$, and fine-tuning steps $K_1$, PGD steps $K_2$, pretrained LDM upscaler set $\Theta_{\mathbb{T}}$, random transformations $\mathbb{T}'$.
2: **for** $i = 0$ to $N - 1$ **do**
3:     Sample $\theta_i$ from $\Theta_i$; sample $\theta_{\mathbb{T}}$ from $\Theta_{\mathbb{T}}$; sample random transformation $g$ from $\mathbb{T}'$.
4:     Copy model weights: $\theta_i' \leftarrow \theta_i$
5:     **for** $j = 0$ to $K_1 - 1$ **do**
6:         Update copyed UNet model weights $\theta_i'$ on $X_p$ according to Eq. 5.
7:     **end for**
8:     **for** $j = 0$ to $K_2 - 1$ **do**
9:         Compute the StyleGuard loss according to Eq. 3, Eq. 4, Eq.6 and Eq. 7.
10:        Optimize protected images $X_p$ using PGD attacks according to Eq. 8.
11:     **end for**
12:     **for** $j = 0$ to $K_1 - 1$ **do**
13:        Update the surrogate model $\theta_i$ according to Eq.9.
14:     **end for**
15: **end for**
16: **Output:** Final protected images $X_p^*$.

---

## 5 Experiments

### 5.1 Experiment Setup

**Datasets** We evaluate StyleGuard's performance on the WikiArt and CelebA datasets to assess its effectiveness against both style mimicry and personalization attacks. Notably, most previous work has focused on only one of these attacks, whereas we are the first to successfully defend against both. The WikiArt dataset comprises 42,129 artworks from 195 different artists, with each piece categorized by its genre (such as impressionism or cubism). For our style mimicry attacks, we randomly selected 40 artists with different art styles and 20 artworks from each artist, using 10 for training and 10 for evaluation. For the CelebA dataset, we randomly select 100 identities, using 10 images per identity to fine-tune the LDM model and another 10 images for evaluation.

**Implementation details** Initially, we use SD v1.4 and SD v1.5 as the substitute models to perturb the images. The attack budget is set as $\frac{8}{255}$ to be same with baselines. The images encoders used to compute the style loss includes VAE, OpenCLIP-ViT-H-14 and OpenCLIP-ViT-bigG-14. The StyleGuard training details and hyperparameters are included in Appendix A.1. During testing, we evaluate two popular fine-tuning methods: DreamBooth and Textual Inversion [11]. For DreamBooth,

we further assess two common settings: full-tuning (Full-FT) and LoRA fine-tuning (LoRA-FT) [14], applied to both SD v2.1 and SD-XL (only LoRA on SD-XL, due to memory constraints). We used the official script from the Diffusers library for DreamBooth fine-tuning. The training details for DreamBooth and Textual Inversion are included in Appendix A.2 and A.3.

**Attack Settings**   We consider three kinds of attacks, including random transformation, DiffPure, and Noise Upscaling. For random transformations, we consider Gaussian noising, center cropping and resizing. For the DiffPure, we use the official code, and use Guided Diffusion Model and DDPM model during training, and use Stable Diffusion XL during evaluation. For Noise Upscaling, we followed the settings outlined by Honig et al. [19]. During the optimization, we generate perturbation on the SD-x4-upscaler [35, 3]. During testing, we evaluate the defense effectiveness using the SD-x2-latent-upscaler [2], which has a different architecture compared to the SD-x4-upscaler (see Appendix A.5 for more details). The number of diffusion steps was set to 30 because large diffusion steps will modify the image content. We also use an online black-box upscaler, Finegrain Image Enhancer (FIE) [1]. For FIE, the upscale factor is set as 2, and the ControlNet scale is set as 0.6.

**Baseline settings**   We compare our method with Glaze [38], Mist [26], Anti-DreamBooth [40], MetaCloak [29] and SimAC [41]. The attack budget are set as $\frac{8}{255}$ for all baselines except Glaze, which uses LPIPS as constraint. The implementation details are included in Appendix A.4.

**Evaluation Metrics**   To evaluate the defense performance for style mimicry, we use Fréchet Inception Distance (FID)[12] and precision [23] as assessment metrics (consistent with Mist[26]) and mimicry success rate [19], which uses human as annotators. To exclude content influence in the style mimicry attack, we use the model trained on clean images to generate 100 paintings in a specific category (e.g., an <sks> painting of a house). We then generate another 100 images from the model trained on style-guarded images using the same prompt and compute the FID and precision between the two sets. For personalization attacks, we use identity match score (IMS) [40] to access the semantic closeness between faces. The metric details are shown in the Appendix A.6.

## 5.2   Experiments Results

**Comparison StyleGuard with Different Methods.**   we evaluate the protection efficacy of our method and baselines under no preprocessing and different transformations. Table 1 shows the evaluation results on the WikiArt using DreamBooth. The baseline trained on clean images with FID (233.78) and Precision (0.60) serves as reference points. It is shown that when there is no attack, all methods can successfully disrupt style mimicry, resulting in increased FID and reduced precision compared to the baseline. The previous method, Mist, demonstrated vulnerabilities to straightforward transformations like cropping and resizing, as well as Gaussian noise, yielding precision scores of 0.46 and 0.48. This weakness stems from its insufficient attention to global features and a lack of consideration for transformations or purifications within its methodology. MetaCloak and SimAC exhibit enhanced robustness to simple transformations because they incorporate these considerations into their pipeline, resulting in high FID scores and low precision scores (0.20 and 0.18 for MetaCloak, 0.06 and 0.04 for SimAC). However, their effectiveness diminishes in the presence of a purifier or upscaler. Our method, StyleGuard, surpasses all existing techniques in both scenarios. It achieves the highest FID and the lowest precision across different transformation settings, demonstrating strong protective efficacy against transformations and purifications. Notably, when purifiers or upscalers are present, our method significantly outperforms the baseline methods. We believe this is because our upscale loss (Eq. 4) can bypass the purifier or upscaler. Additionally, our human evaluation results corroborate these findings, shown in Figure 4 (see Appendix A.7 for human evaluation details).

**The Importance of Different Loss Functions**   The efficacy of various loss functions in safeguarding against style mimicry is visualized and quantitatively analyzed in Figure 3 and Table 1. Figure 3 illustrates the influence of different loss functions on image quality and the robustness of protection. Although using denoise loss and style loss can result in a decrease in image quality for unprotected images (Figure 3 (2)), it is still vulnerable to Noise Upscale, as shown in Figure 3 (3), in which the image quality is improved. The integration of upscale loss significantly improves the protection resilience, as evidenced in Figure 3 (4). This underscores their collective contribution to a more robust defense mechanism, ensuring that style mimicry is effectively countered even when purification attacks exist. The quantitative results in Table 1 further validate the effectiveness of the loss functions.

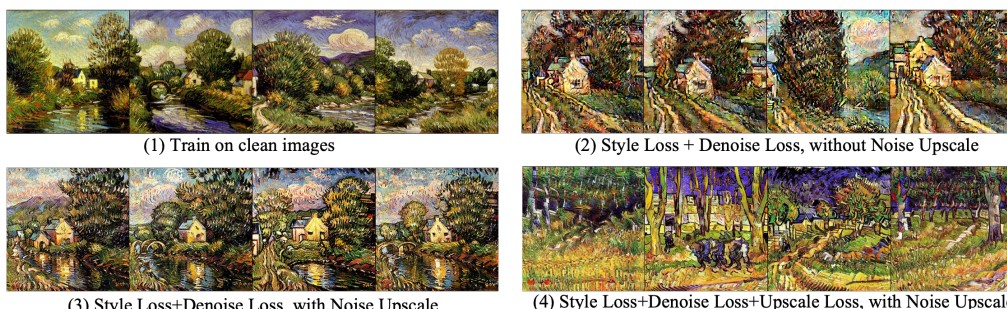

|  |  |
|---|---|
| (1) Train on clean images | (2) Style Loss + Denoise Loss, without Noise Upscale |
| (3) Style Loss+Denoise Loss, with Noise Upscale | (4) Style Loss+Denoise Loss+Upscale Loss, with Noise Upscale |

Figure 3: Visualizing the effects of different loss functions. It is shown that only using denoise loss and style loss cannot defend the Noise Upscale well, as shown in (3). With the upscaler loss, the image quality significantly decreases even with Noise Upscale, as shown in (4).

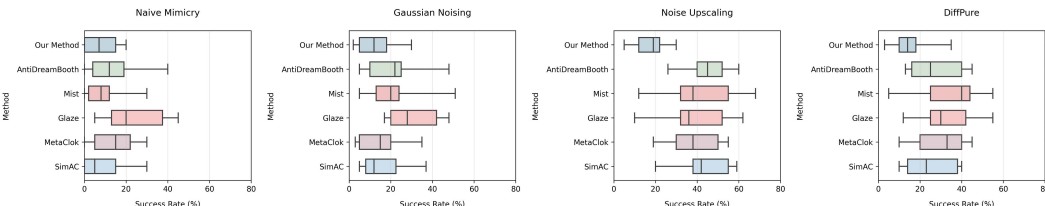

Figure 4: Evaluation results of mimicry success rates by human evaluators. We asked users to compare generated images based on clean and protected training images using the question: "Based on the image style and quality, which image better fits the reference samples?" A lower mimicry success rate indicates stronger perturbation noises affecting the image quality.

StyleGuard, which incorporates denosing loss, style loss, and upscale loss, achieves the highest FID and the lowest Precision, indicating that the image quality is lower than using single loss.

**Textual Inversion Results**    Textual Inversion optimizes only a small set of new token embeddings that can later be appended to prompts to imitate the target style/image without fine-tuning the model. We use the official code for the textual inversion from the diffusers package and use the SD v2-1-base as the text-to-image model. The results of the experiment are shown in Table 4 in the Appendix. Experiments show that StyleGuard's denoising loss is more effective against textual inversion. This is because Textual Inversion does not modify model weights and the adversarial noise directly corrupts the semantic alignment between learned tokens and the style. The style loss can destroy the quality of the generated images further, as shown in the second-to-last line in Table 4. It is shown that StyleGuard remains effective even when attackers use a different method, including DreamBooth and Textual Inversion, whereas traditional defenses (e.g., Glaze) fail.

**Transferability on different models**    High transferability method is more practical for real-world applications. Table 2 compares the transferability of Anti-DreamBooth (above the slash) and Style-Guard (under the slash). We calculated the ratio of FID scores (%) for images generated on evaluation models and substitute models. The higher the ratio, the better the transferability. The results demonstrate that StyleGuard has better transferability across different SD models than Anti-DreamBooth. This is because, compared to Anti-DreamBooth, which only uses denoising loss, StyleGuard incorporates style loss. This addition can perturb global style-related features that are independent of the model parameters. We also evaluate the protection results on a black-box online upscaler, FIE. As shown in Figure 8 in the Appendix, when using FIE for purification, the style of the image is more similar to Van Gogh's style compared to images without purification. However, it still does not match the quality of images generated from clean inputs.

**Experiment Results on LoRA**    To evaluate cross-model transferability under different fine-tuning settings, we apply LoRA-based training to SD models optimized with DreamBooth loss (3). Quanti-

Table 1: Comprehensive evaluation of text-to-image protection methods under different transformations for DreamBooth on the WikiArt Dataset. Metrics reported are FID↑ (higher better) and Precision↓ (lower better). The best data are shown in bold, and the second runners are in gray.

| Method | No Prep. | | Crop+Resize | | Gauss. Noise | | DiffPure | | Noise Up. | |
|---|---|---|---|---|---|---|---|---|---|---|
| | FID | Prec. | FID | Prec. | FID | Prec. | FID | Prec. | FID | Prec. |
| No Protect | 233.78 | 0.60 | 275.41 | 0.65 | 238.25 | 0.62 | 237.89 | 0.68 | 236.58 | 0.60 |
| Glaze | 333.89 | 0.15 | 315.22 | 0.40 | 340.10 | 0.35 | 318.94 | 0.30 | 312.73 | 0.60 |
| Mist | 382.50 | **0.00** | 295.28 | 0.46 | 275.77 | 0.48 | 290.45 | 0.42 | 256.65 | 0.45 |
| AntiDB | 327.01 | 0.05 | 310.88 | 0.25 | 322.15 | 0.20 | 305.74 | 0.35 | 293.14 | 0.50 |
| MetaCloak | 382.00 | 0.05 | 362.52 | 0.20 | 355.26 | 0.18 | 318.87 | 0.25 | 295.20 | 0.40 |
| SimAC | 407.40 | **0.00** | 365.47 | 0.06 | 380.45 | 0.04 | 290.15 | 0.38 | 284.52 | 0.45 |
| $L_{denoise}$ | 348.15 | 0.03 | 355.77 | 0.30 | 362.42 | 0.15 | 358.89 | 0.12 | 310.54 | 0.20 |
| $L_{denoise+style}$ | 389.33 | 0.01 | 382.45 | 0.25 | 380.21 | 0.08 | 385.77 | 0.06 | 375.92 | 0.10 |
| StyleGuard | **428.70** | **0.00** | **405.31** | **0.05** | **420.74** | **0.02** | **418.33** | **0.03** | **401.80** | **0.00** |

tative results demonstrate that our method achieves stronger performance on SD-XL (FID: 464.45, Precision: 0.00) compared to SD-v2-1 (FID: 366.78, Precision: 0.16), though both scenarios significantly outperform prior approaches. We hypothesize that the weaker protection efficacy on SD-v2-1 stems from architectural differences in parameter adaptation during LoRA fine-tuning. Specifically, SD-v2-1 may undergo updates concentrated in fewer layers, particularly those less sensitive to adversarial perturbations, resulting in a smaller effective attack surface (see Figure 6 in Appendix for visual examples).

**Evaluation Results on Personalization Attacks**   We also evaluate the effectiveness of our method in defending against personalization attacks (see Appendix A.8 for implementation details). As shown in Figure 7 and Table 5 in the Appendix, we successfully defend against the personalization attack by DreamBooth, even when Noise Scaling is used to preprocess the face images. We think this is because the style loss can not only disrupt the style-related features but also the identity-related features, which is consistent with the findings of StyleGAN [21].

Table 2: Cross-model transferabilities for AntiDreamBooth and StyleGuard.

| Surrogate ↓ | Evaluation model | | |
|---|---|---|---|
| | SD v1.4 | SD v1.5 | SD v2.1 |
| SD v1.4 | 100.0/100.0 | 85.5/96.5 | 76.5/92.4 |
| SD v1.5 | 84.8/96.2 | 100.0/100.0 | 73.5/92.5 |
| SD v2.1 | 73.5/89.2 | 76.4/92.5 | 100.0/100.0 |

Table 3: Transferability to different fine-tuning methods.

| Finetuning Method | LoRA/SD v2-1 | | LoRA/SD XL | |
|---|---|---|---|---|
| | FID ↓ | Prec. ↑ | FID ↓ | Prec. ↑ |
| Clean (Baseline) | 210.45 | 0.75 | 215.60 | 0.72 |
| AntiDreamBooth | 285.20 | 0.45 | 365.80 | 0.28 |
| MetaCloak | 320.85 | 0.28 | 435.60 | 0.22 |
| SimAC | 375.90 | 0.20 | 445.75 | 0.12 |
| Ours | 366.78 | 0.16 | 464.45 | 0.00 |

# 6 Limitation and Conclusion

This work introduces StyleGuard, a novel and robust anti-mimicry method designed to protect artists from unauthorized style mimicry in text-to-image diffusion models. By optimizing style-related features in the latent space, StyleGuard effectively disrupts the extraction of correct style features, making it difficult for attackers to replicate the original artistic style. Extensive experiments on the WikiArt and CelebA-HQ datasets show that styleGuard exhibits strong cross-model transferability, outperforming existing methods in terms of protection efficacy. Our approach also demonstrates superior robustness against data transformations, including the state-of-the-art DiffPure and Noise Upscaling. This work addresses critical challenges in intellectual property protection in the digital art domain, providing artists with a powerful tool to safeguard their unique styles from unauthorized exploitation. However, experiments indicate that StyleGuard is less effective with commercial upscalers and LoRA-based fine-tuning methods. Future work could explore extending StyleGuard to other customization methods and more complex purification methods.

## Acknowledgment

This work was supported in part by the Hong Kong Research Grants Council's (RGC) General Research Fund (GRF) under Grant PolyU 15201323.

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

# A    Technical Appendices and Supplementary Material

## A.1    StyleGuard Training Details

We generate the perturbations using eight NVIDIA 3090 GPUs. The fine-tuning steps, denoted as $K_1$, are set to 3, while the PGD steps, denoted as $K_2$, are set to 6. The total training steps $N$ are set to 100. The target image is randomly selected from a different art genre. The PGD budget is configured to $\frac{8}{255}$, and the PGD step size is set to 0.005. Additionally, the weight of upscale loss ($\eta$) is set to 1, and the weight of the style loss ($\lambda$) is set to 10. The optimization time for each image is approximately 20 minutes, compared to 15 minutes for MetaCloak [29] and 40 minutes for SimAC [41]. This extended time for SimAC is due to the additional steps required to select time steps for the denoising error loss, which we find provide minimal benefit for the defense.

## A.2    DreamBooth Training Details

For the training of DreamBooth, we use a learning rate of 5e-6, with the prior reservation loss weight set to 1.0. For the style mimicry attack, the customized prompt is set as "an <sks> painting" and the prior prompt was set as "a painting". For the personalization attack, the customized prompt is set as "a photo of <sks> person", and the prior prompt was set as "a photo of person". In the original DreamBooth paper, they use 5 images for the personalization. However, we found that using only 5 images per artist was insufficient for generating high-quality mimicry. Therefore, we opted for 10 images per artist and trained for 1000 epochs. For LoRA fine-tuning, the dimension of the LoRA update matrices is set as 4. We use mixed precision of bf16 to save memory. It takes about 20 minutes for the full fine-tuning and 4 minutes for the LoRA fine-tuning.

## A.3    Textual Inversion Implementation Details and Experiment Results

Textual Inversion is a technique for learning and replicating new visual concepts (e.g., artistic styles, objects, or aesthetics) in a pretrained text-to-image diffusion model (such as Stable Diffusion) without fine-tuning the model's weights. Instead, it optimizes only a small set of new token embeddings that can later be appended to prompts to imitate the target style/image. The method introduces new (placeholder) text tokens (e.g., "sks_style") and optimizes their embeddings (vectors in the text encoder's space) to represent the target visual style. These tokens can later be inserted into prompts (e.g., "A painting in the style of sks_style") and will guide the model to generate images resembling the trained style. We use the official code for the textual inversion and use the SD v2-1-base as the text-to-image model. In our experiment, we find that StyleGuard's denoising loss is most effective against Textual Inversion (Table 4) while the style loss is also important to destroy the quality of generated images. This is because Textual Inversion does not modify model weights and the adversarial noise directly corrupts the semantic alignment between learned tokens and the style.

Moreover, StyleGuard's perturbations remain effective even when attackers use different mimicry methods (Dreambooth, Textual Inversion, etc.), whereas traditional defenses (e.g., Glaze) fail. Previous work, like MetaCloak and SimAC, are robust to simple transformations like Crop+Resize and Gaussian Noise. However, they are relatively vulnerable to attacks such as DiffPure and Noise Upscaling. When there is DiffPure and Noise Upscaling, our methods achieve the highest FID and the lowest precision. We think this is because the StyleGuard methods consider a variety of purifiers and upscalers during training. However, when there is no purifying measure, our method performs slightly worse than SimAC in textual inversion, because SimAC adds the step of selecting diffusion timesteps. Although SimAC can further reduce the quality of the image, it will also increase the optimization time.

## A.4    Baseline Implementation Details

In this section, we outline the baseline settings for our comparisons with several protection methods: Glaze, Mist, Anti-DreamBooth, MetaCloak, and SimAC. To ensure a fair evaluation, we maintain a consistent perturbation budget of $p = 8/255$ for all methods except Glaze. Evaluating Glaze under this specific budget presents challenges due to Glaze utilizes LPIPS for its image similarity metric, which does not constrain the $L^\infty$ norm. Consequently, we implement Glaze by our own according to the Glaze paper. Our observations indicate that images processed with Glaze appear equally or less perturbed compared to those processed with Mist and Anti-DreamBooth.

Table 4: Comprehensive evaluation of text-to-image protection methods under different transformations for Text Inversion. Metrics reported are FID↑ (higher better) and Precision↓ (lower better).

| Method | No Preprocess | | Crop+Resize | | Gaussian Noise | | DiffPure | | Noise Upscale | |
|---|---|---|---|---|---|---|---|---|---|---|
| | FID | Prec. | FID | Prec. | FID | Prec. | FID | Prec. | FID | Prec. |
| No Protection | 237.56 | 0.9 | 280.63 | 0.88 | 241.20 | 0.87 | 239.87 | 0.92 | 255.31 | 0.85 |
| Glaze | 249.43 | 0.84 | 263.05 | 0.72 | 285.11 | 0.75 | 262.33 | 0.65 | 285.47 | 0.71 |
| Mist | 454.39 | 0.02 | 297.90 | 0.80 | 422.17 | 0.10 | 301.25 | 0.78 | 294.60 | 0.75 |
| AntiDB | 371.12 | 0.22 | 327.88 | 0.44 | 353.45 | 0.31 | 266.54 | 0.52 | 260.13 | 0.58 |
| MetaCloak | 416.25 | 0.04 | 380.52 | 0.20 | 398.76 | 0.12 | 315.87 | 0.68 | 308.42 | 0.72 |
| SimAC | **465.82** | 0.02 | 370.87 | 0.18 | **441.35** | 0.08 | 340.15 | 0.25 | 335.79 | 0.30 |
| $L_{\text{denoise}}$ | 382.15 | 0.06 | 385.67 | 0.08 | 375.42 | 0.22 | 368.79 | 0.19 | 362.84 | 0.25 |
| $L_{\text{denoise+style}}$ | 410.33 | 0.04 | 392.45 | 0.05 | 403.21 | 0.12 | 395.67 | 0.10 | 388.92 | 0.15 |
| StyleGuard | 428.57 | 0.00 | **398.21** | 0.05 | 419.84 | 0.08 | **412.33** | **0.04** | **425.76** | **0.07** |

Next, we specify the hyperparameters utilized for replicating each protection method.

**Glaze**  Due to the lack of access to a shared codebase from the Glaze authors, we implemented Glaze independently. The LPIPS distance is computed using the VGG model. In Figure 5, we display examples of images generated by Glaze. The results indicate that Glaze produces images that are a mixture of the target and reference styles.

**Mist**  We conducted our evaluation of Mist following the methodology [26]. The parameters set for this evaluation include a PGD perturbation budget of $p = 8/255$, with $N_{PGD} = 100$ iterations and a PGD step size of $\alpha = 1/255$. The target image used for this evaluation is denoted as $T = $ Target Mist, as illustrated in Figure 5.

**Anti-DreamBooth**  Anti-DreamBooth [40] is tailored to counter DreamBooth fine-tuning. We adapted their approach for our setting focused on style mimicry, retaining their hyperparameters where feasible. We established the following parameters: the number of iterations $N = 50$, PGD perturbation budget $p = 8/255$, PGD step size $\alpha = 5 \times 10^{-3}$, and the number of PGD steps per ASPL iteration $N_{PGD} = 6$. The loss $L_{Finetune}$ is minimized within the vanilla fine-tuning framework over 300 training steps.

**MetaCloak**  We implement MetaCloak [29] using the original setting, with a surrogate pool of 5 diffusion models ($M = 5$). The transformation set $\mathcal{T}$ includes Gaussian filtering (kernel=7), random flips, and center cropping. We use Adam optimizer with $\beta = 10^{-4}$ and $C = 4000$ crafting steps. The denoising-error maximization loss combines with EOT to ensure transformation robustness.

**SimAC**  Following [41], we implement their feature interference loss with adaptive timestep selection. The perturbation budget matches other baselines ($\ell_\infty = 8/255$), and we use their recommended layer weights (9-11) for high-frequency feature disruption. The hyperparameters are same with the official implementation, with a number of training epochs of 50 and each epoch include 3 steps for surrogate model training and 9 steps for the PGD attacks. For the timestep search, the maximum greedy search steps is set as 50.

### A.5 Attack Implementation Details

**The implementation details of noise upscaling.**  Previous work have found that upscaling images can purify adversarial images [33]. Recent work improve this by first applying Gaussian noises and then upscales the noisy image [19]. However, [19] does not specify the model they used for upscaling the images. In our experiment, we train on the SD-x4-upscaler model and then test on the SD-x2-latent-upscaler model. We select these two models because they have different architectures, therefore we can test the transferability of our methods. Specifically, the SD-x4 first encodes the image through a VAE encoder, producing latent features of shape $[B, 4, H, W]$. It then combines a downscaled image with the latent features, resulting in a tensor of shape $[B, 7, H, W]$. In contrast, the SD-x2-latent directly utilizes the latent features.

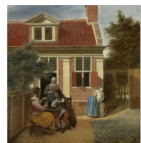

Reference Image

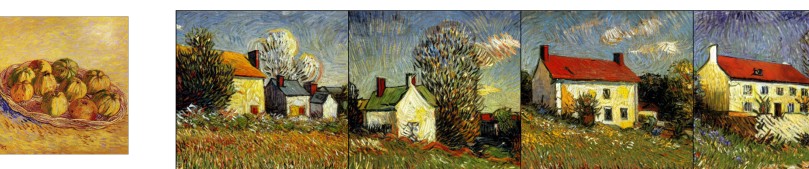

Image finetuned on the clean image, with prompt "an sks painting of a house"



Target Image used in Glaze

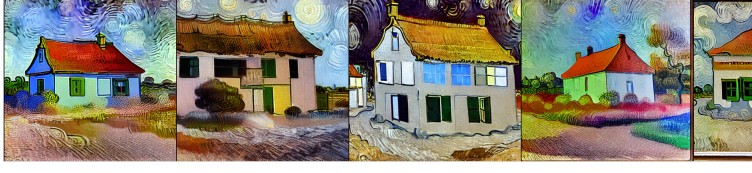

Glaze, using paintings from Artist Rembrandt as target, with prompt "an sks painting of a house"

Target Image used in Mist

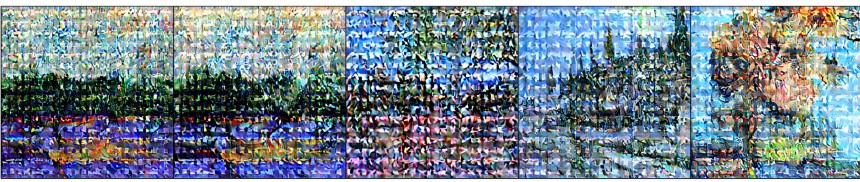

Mist, with prompt "an sks painting of a house"

Figure 5: Visualization of Glaze and Mist. For Glaze, we use paintings from Van Gogh as the reference and paintings from Rembrandt as the targets. The results indicate that Glaze produces images that are a mixture of the target and reference styles. For Mist, we use a periodic image as the target, according to the original paper.

## A.6 Evaluation Metrics

For style mimicry attack, we use three different metrics, FID, precision and success rate.

- **FID**. The Fréchet Inception Distance (FID) measures the statistical similarity between real and generated images by comparing their feature distributions in Inception-v3's latent space. High FID indicates that generated images deviate from the real data manifold, making style imitation harder. Unlike Precision (which measures mode coverage), FID penalizes unnatural artifacts, making it ideal for measuring adversarial disruption.

- **Precision**. The precision metric is computed to evaluate the quality of generated images by assessing their fidelity to the target data manifold. Following the methodology proposed by [23], we implement a manifold-based approach for precision computation. Given two sets of feature embeddings—reference features Given feature embeddings $\mathbf{F}_r$ (real data) and $\mathbf{F}_g$ (generated data)—we proceed as follows: For each real sample $\mathbf{f}_r \in \mathbf{F}_r$, compute its $k$-nearest neighbor radius $R_r(\mathbf{f}_r)$ in $\mathbf{F}_r$.

$$\text{Precision} = \frac{1}{|\mathbf{F}_g|} \sum_{\mathbf{f}_g \in \mathbf{F}_g} \mathbb{I}\left(\exists\, \mathbf{f}_r \in \mathbf{F}_r : \|\mathbf{f}_g - \mathbf{f}_r\|_2 \leq R_r(\mathbf{f}_r)\right) \tag{10}$$

where $\mathbb{I}(\cdot)$ is the indicator function. Higher precision indicates better coverage of real data modes. To exclude the influence of image content in the style mimicry attack, we first use the fine-tuned model on a clean image to generate 100 new paintings in a specific category (e.g., an SKS painting of a house). We then use the fine-tuned model on the style-guarded images to generate another set of 100 images with the same prompt. These two sets of images are then used to compute the precision.

- **Mimicry Success Rate**. Mimicry success rate employs human annotators in a pairwise comparison protocol to evaluate generated images from unprotected inputs versus those

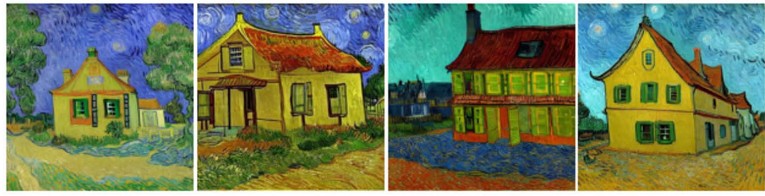

(a) DreamBooth-LoRA, finetune SD v2.1 on unprotected image

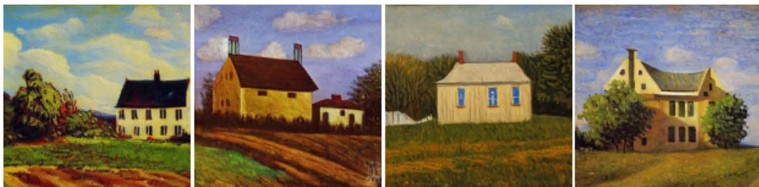

(b) DreamBooth-LoRA, finetune SD v2.1 on protected image

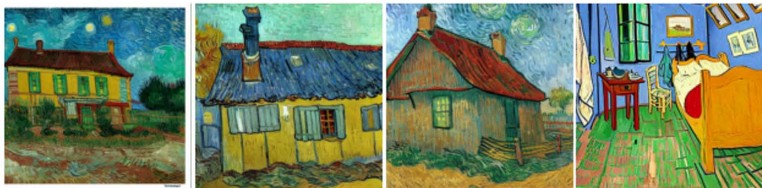

(c) DreamBooth-LoRA, finetune SD XL on protected image

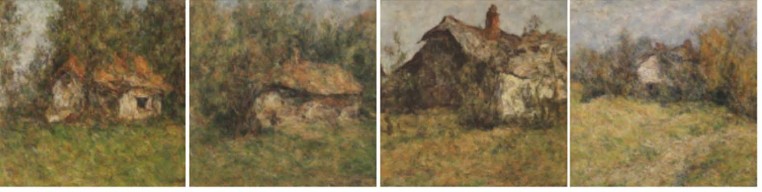

(d) DreamBooth-LoRA, finetune SD XL on unprotected image

Figure 6: Comparison of images generated from unprotected and protected images using LoRA methods. We utilized the DreamBooth loss to fine-tune the SD v2.1 and SD XL models. Our findings reveal that our method significantly reduces image quality for the SD XL model. Although the image quality degradation for the SD v2.1 model is less pronounced, there is still a notable change in style.

from protected images.

$$\text{success rate} = \frac{1}{N_p * N_a} \sum_{prompt} \sum_{annotator} [\text{robust mimicry preferred over unprotected mimicry}]$$

(11)

A perfectly robust mimicry method would thus obtain a success rate of 50%, indicating that its outputs are indistinguishable compared with the unprotected method. In contrast, a severely restricted protection would result in success rates around 0% for robust mimicry methods, indicating that mimicry on top of protected images always yields worse outputs. In the experiment, we use 5 different annotators. Each annotator needs to compare 10 image pairs for each transformation and protection method.

For the personalization attack, we use identity match score (IMS), which computes the similarity between the embedding of generated face images and an average of all reference images. We use VGG-Face and CLIP-ViT-base-32 as embedding extractors to extract face features and employ the cosine similarity.

## A.7 Human Evaluation Results on the Style Mimicry

To further evaluate the success rate under different transformation settings, we asked human annotators to compare images generated by models fine-tuned on unprotected versus protected images. Annotators assessed these two sets of images based on style and quality, selecting which set exhibited better quality. Thus, a successful mimicry attack would yield a success rate of nearly 50%, indicating competitiveness with images trained on clean samples.

We employed five different annotators, each tasked with comparing ten image pairs for 4 transformations and 6 protection methods. Metrics were computed using Equation 11. The results are presented in Figure 4, which shows that when purifications such as DiffPure and Noise Upscaling are applied, our method's success rate is significantly lower than baseline methods.

## A.8 Evaluation Results on the Personalization

We evaluated the effectiveness of our method in defending against personalization attacks, where an adversary attempts to generate customized face images using a fine-tuned model. As shown in Figure 7, we successfully defend against the personalization attack by DreamBooth, even when Noise Scaling is used to preprocess the face images. The top row, $X_{clean}$, represents the original, unaltered input, while the images labeled $X_{p*}$ correspond to the perturbed images generated by StyleGuard.

For quantitative analysis, we selected 100 identities from the CelebA dataset, choosing 10 images for each identity to fine-tune the LDM using the DreamBooth loss function. The PGD budget was set to 16/255. The success of the defense is measured by the Identity Matching Score [40], which computes the cosine distance between the generated images and the average face embedding of the user's clean image set using the VGG-Face and CLIP-ViT-base-32. A lower ISM indicates that the model cannot reproduce images of the same identity. The results are shown in Table 5. It is evident that when no purifications are applied, all methods achieve successful protection against anti-personalization. The best method, MetaCloak, achieves an $IMS_{CLIP}$ of 0.662 and an $IMS_{VGG}$ of -0.051. However, when noise upscaling is introduced, the protective strength of previous methods significantly decreases. In contrast, our method continues to effectively mitigate personalization attacks.

Table 5: Comparison of Identity Matching Score on the anti-personalization for different methods. The lower the IMS, the stronger the protection is. When no purifications are applied, all methods achieve successful protection against anti-personalization. However, when noise upscaling is introduced, the protective strength of previous methods significantly decreases. In contrast, our method continues to effectively mitigate personalization attacks.

| Method | $IMS_{CLIP}$ | $IMS_{VGG}$ | $IMS_{CLIP}$ with UpScale | $IMS_{VGG}$ with Upscale |
|---|---|---|---|---|
| Clean | 0.814 | 0.432 | 0.805 | 0.429 |
| Anti-DreamBooth | 0.695 | -0.012 | 0.725 | 0.379 |
| SimAC | 0.675 | -0.022 | 0.780 | 0.373 |
| MetaCloak | **0.662** | **-0.051** | 0.748 | 0.354 |
| Our Method | 0.680 | -0.039 | **0.685** | **0.120** |

## A.9 Evaluation over Online Black-box Upscaler

We use an online black-box upscaler, Finegrain Image Enhancer (FIE) [1], to further evaluate our method. For FIE, we use the default setting, with the upscale factor set as 2 and the ControlNet scale set as 0.6. The Gaussian noise standard deviation is set as 0.1. Figure 1 shows the results of fine-tuning using a Van Gogh-style image. The first row is the result of fine-tuning using a clean image. The second row is the result of fine-tuning using a protected image after purification and then using Dreambooth. When using FIE for purification, the style of the image is somewhat similar to Van Gogh's style, but it is still not as good as the fine-tuning result of the clean image. We believe that this is because FIE has a high degree of denoising, which can purify noise to a certain extent, but will also modify the content of the image to a certain extent, such as the details of the brushstrokes, changes in lines, etc., thus causing the style of the fine-tuned image to change.

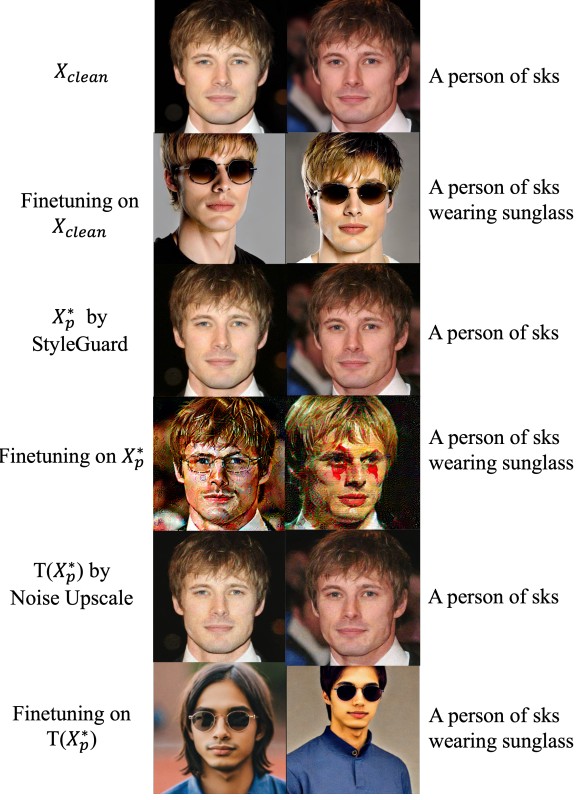

| | | |
|---|---|---|
| $X_{clean}$ | | A person of sks |
| Finetuning on $X_{clean}$ | | A person of sks wearing sunglass |
| $X_p^*$ by StyleGuard | | A person of sks |
| Finetuning on $X_p^*$ | | A person of sks wearing sunglass |
| T($X_p^*$) by Noise Upscale | | A person of sks |
| Finetuning on T($X_p^*$) | | A person of sks wearing sunglass |

Figure 7: Defense against Personalization Attack by Dreambooth using StyleGuard.

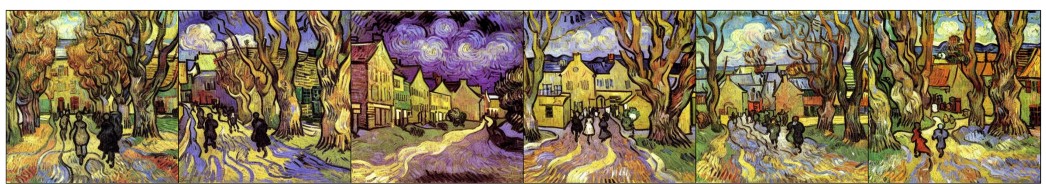

SD model trained with clean images, with prompt "a <sks> painting of people waking on the street"

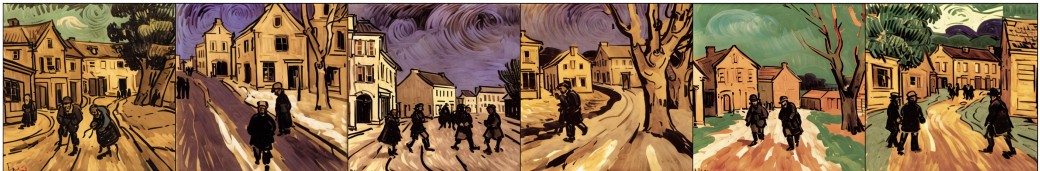

SD model trained with protected images (purified with FIE), with prompt "a <sks> painting of people waking on the street"

Figure 8: Visualization over Online Black-box Upscaler FIE. When using FIE for purification, the image style is more similar to Van Gogh's than without purification, but it is still not as good as the fine-tuning result from the clean images.

## B. Additional Demonstration of Visual Examples

### B.1 Compare the Clean, Protected and Upscaled Images

To demonstrate the effectiveness of our method, Figure 9 compares original, clean images without any processing with protected versions after applying our protection methods. and the noise-upscaled images that the adversary applies Noise Upscale with a different version of the upscale model (SD-x2-latent-upscaler) from the training stage to the protected images. We have made some findings as follows. First, it is shown that the noise introduced by our method is very small and does not affect the image quality. Second, Noise Upscale can better restore image details, such as the face in the sixth row. We think this may be because Van Gogh's image appears in the Upscaler training dataset. However, for some parts of the image style, Noise Upscale cannot be restored well, such as the sky in the first row and the grass in the fifth row, which become even blurrier after Upscale. We think this is because these images may not in the training images of the Upscale model.

### B.1 Compare the Images Trained on Clean Images and Protected Images

Figure 10 and Figure 11 compare the results of style mimicry on clean and protected images with the StyleGuard protection. For StyleGuard, we generate perturbations using SD1.4 and SD x4 upscaler. During the test, we first apply the Noise Upscale using the SD x2 upscaler and then train the SD1.5 model on the protected paintings. With protection, the quality of the protected image decreases significantly, and its style changes from the original image.

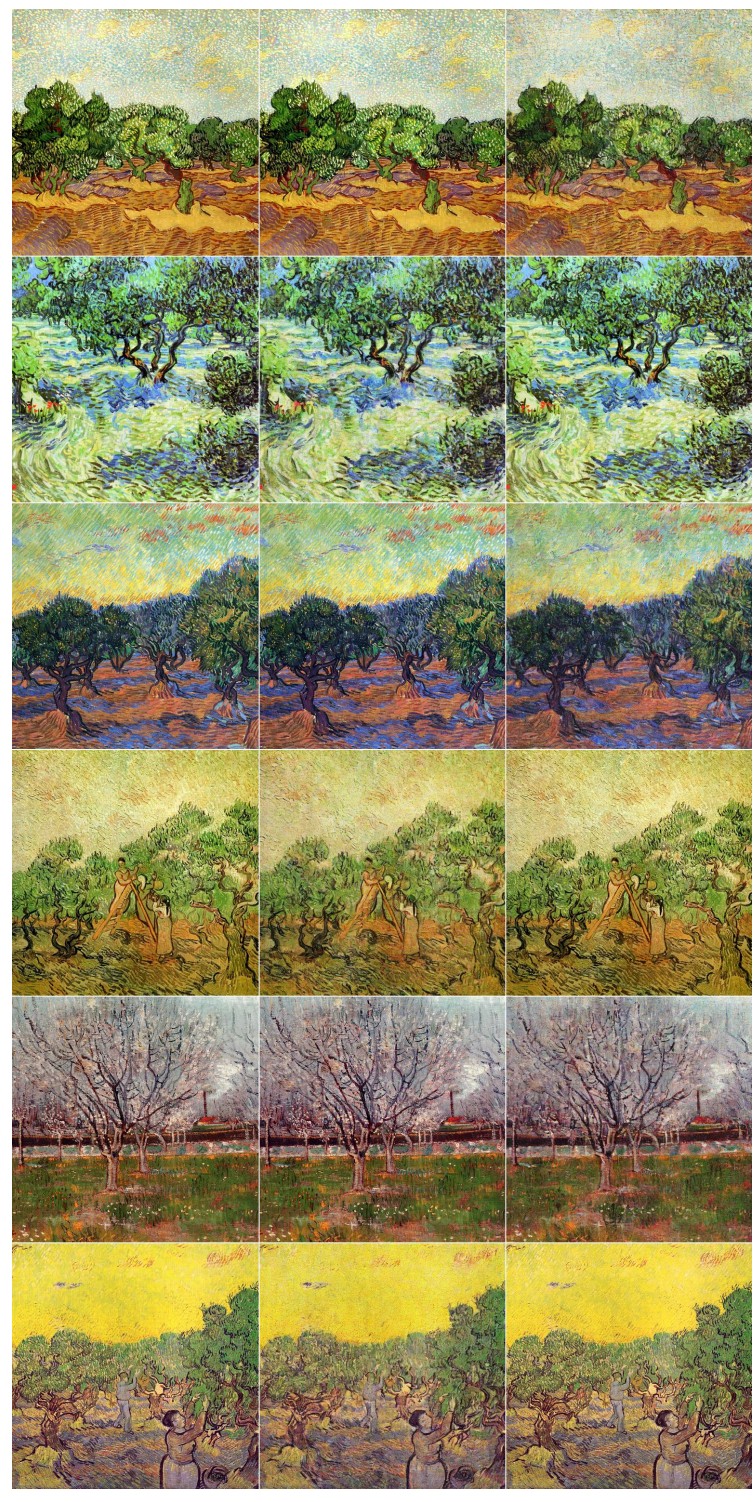

Figure 9: Visual comparison between (a) clean/original images (left column), (b) protected images (middle column), and (c) Noise Upscaled results (right column). Each row shows the same image processed through different pipeline stages.

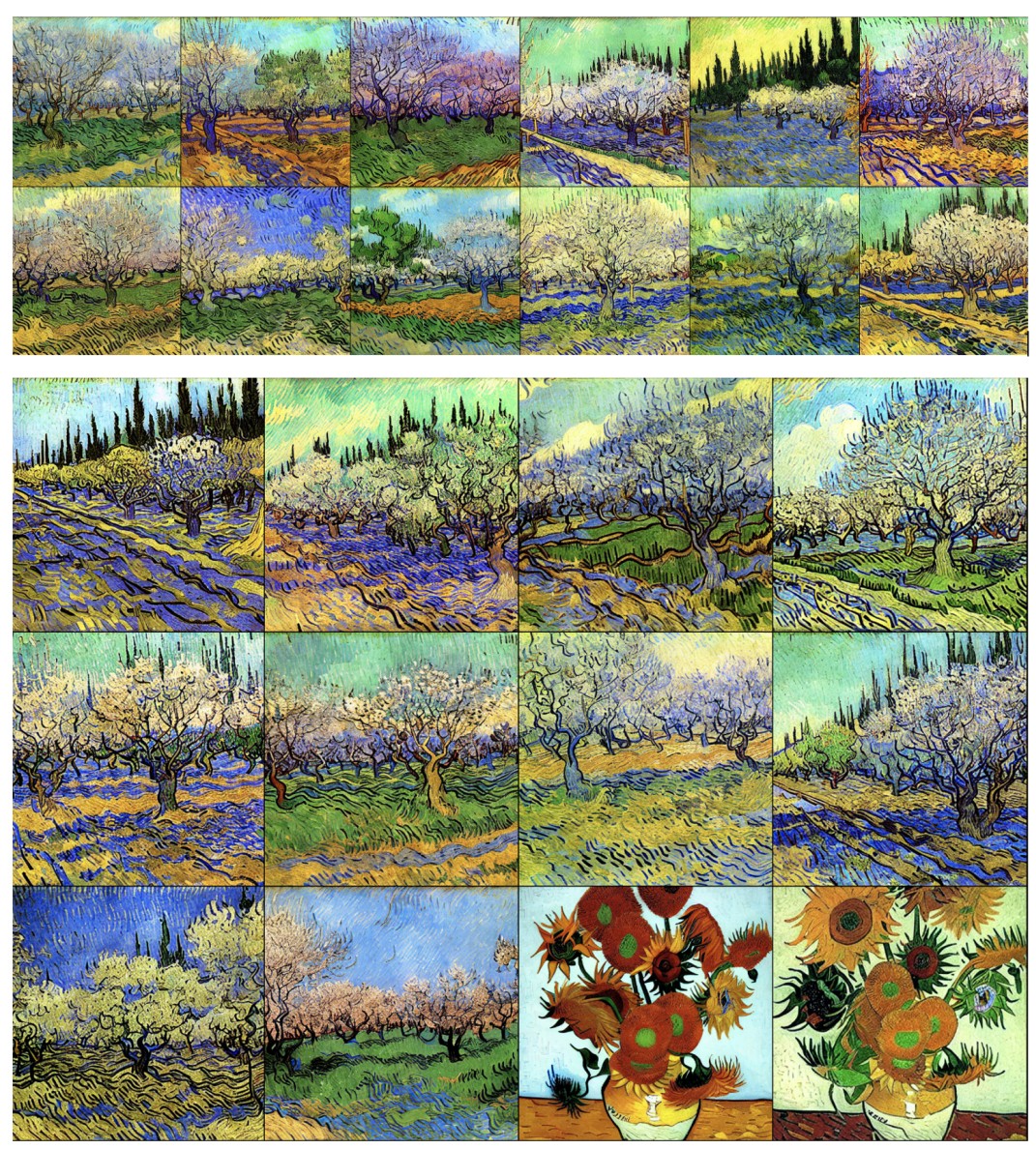

Figure 10: The results of style mimicry on clean images without any protection. We train the SD v1.5 model on Van Gogh's paintings.

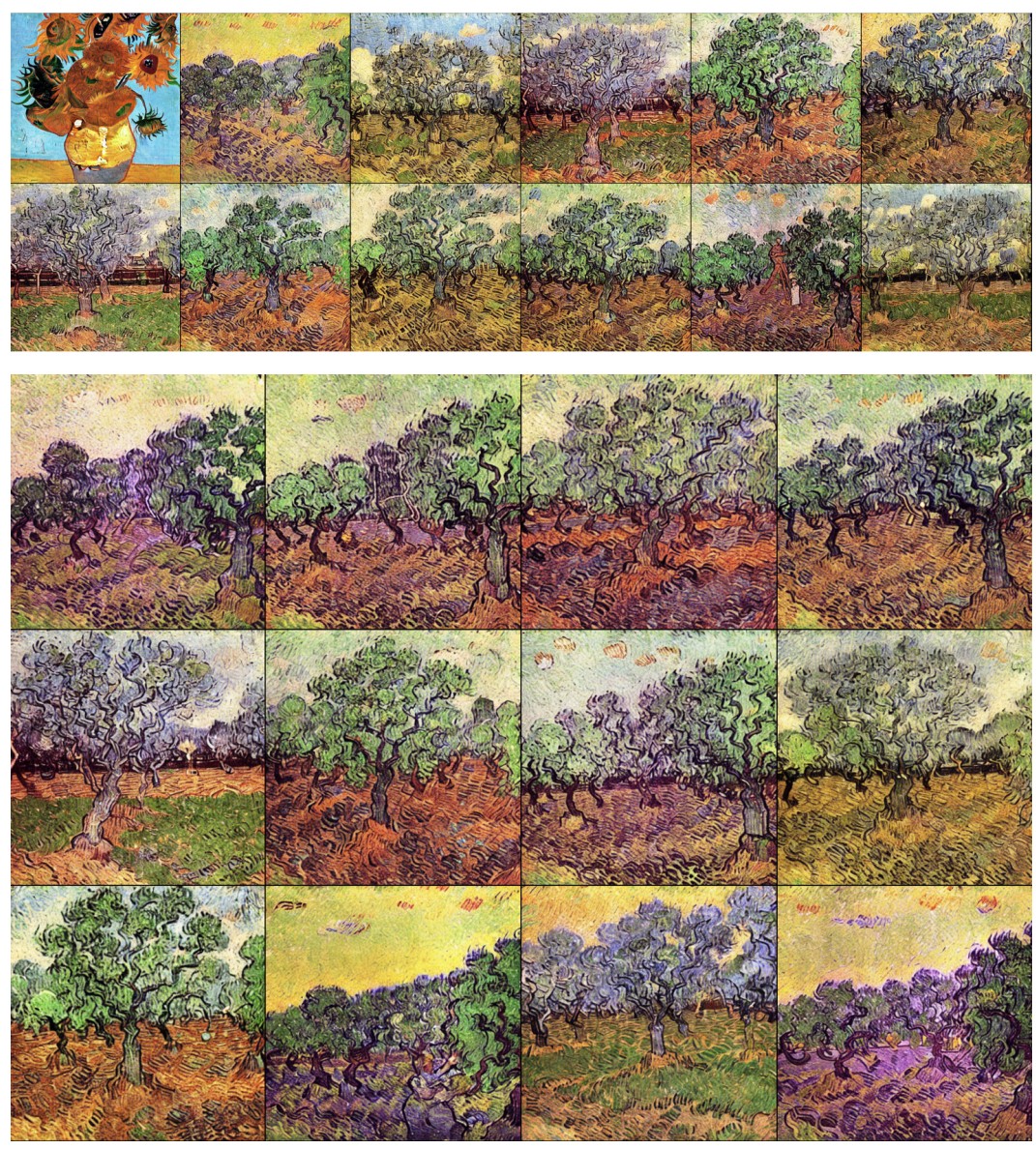

Figure 11: The results of style mimicry on protected images with the StyleGuard protection. For the StyleGuard, we generate perturbations using SD1.4 and SD x4 upscaler. During the test, we first apply the Noise Upscale using the SD x2 upscaler and then train the SD1.5 model on the protected paintings.

