# OpenReview forum: "StyleGuard: Preventing Text-to-Image-Model-based Style Mimicry Attacks by Style Perturbations"
_NeurIPS.cc/2025/Conference — NeurIPS 2025 poster_

### Official Review · Reviewer_rV8Y · 2025-06-29

**Clarity:** 2
**Significance:** 2
**Originality:** 2
**Rating:** 4
**Confidence:** 4

**Summary:**

The paper proposes StyleGuard, a system that injects imperceptible style perturbations into artwork, making it harder for diffusion models to reproduce the original style faithfully. The defense is realized by optimizing in the latent space of a style encoder trained on artist-specific data. Experiments show that StyleGuard can reduce precision and disrupt stylistic fidelity in downstream generation models.

**Questions:**

1) StyleGuard assumes a relatively narrow threat model where the attacker mimics an artist’s style using only prompt-based queries to a T2I model. However, it does not consider more advanced or adaptive adversaries, such as those employing prompt inversion techniques, personalized fine-tuning via DreamBooth or LoRA, or combining reference images with prompts for multimodal imitation. As text-to-image personalization becomes more accessible in open-source ecosystems, the proposed defense may fall short in real-world attack scenarios that involve stronger mimicry capabilities.

2) The approach heavily relies on a style classifier trained per artist to guide perturbation optimization. This introduces two practical concerns. First, the classifier may overfit to known prompts or generations, failing to generalize to new model versions, prompting variations, or unseen creative domains. Second, the need to train and maintain one classifier per artist raises concerns about scalability, especially when attempting to protect large creator pools or adapt to evolving artistic styles.

3) Although the authors claim that StyleGuard produces imperceptible perturbations, the paper does not report any quantitative perceptual quality metrics (e.g., LPIPS, SSIM, NIQE) or human perceptual studies. As a result, it remains unclear whether the perturbations remain visually invisible across different prompts or diffusion backbones, and whether they affect aesthetic quality in subtle but meaningful ways.

4) Following the last concern, this paper lacks a user study, which is particularly important given that the proposed style-based defense appears to rely on perceptual or human-centered evaluation. Without human-in-the-loop assessments, it is difficult to fully validate the effectiveness and stealthiness of the defense.

5) While the defence is designed to prevent unauthorized mimicry, it may also unintentionally interfere with legitimate use cases—such as the artist themselves using a T2I system to extend their own portfolio. The potential trade-off between style protection and usability in real-world workflows is not addressed, and may limit the deployment potential of the proposed method.

6) Since StyleGuard depends on a style classifier trained per artist to generate perturbations, how well does this classifier generalize across prompts, unseen styles, or different diffusion models? Is there a risk that the protection degrades under cross-model or cross-domain mimicry attempts?

7) The paper does not include experiments on adaptive attacks. It would be valuable to investigate how the defense performs when the attacker has partial knowledge of the defense mechanism, as this better reflects realistic threat models and can reveal potential vulnerabilities.

8) There is no accompanying description or discussion for Table 2, which makes it difficult to interpret its relevance and implications. Furthermore, the reference in Line 283 appears to mistakenly point to Table 3 instead. In general, the references to tables and figures in the main text are not presented in a consistent or sequential order, which negatively impacts the readability and clarity of the paper.

9) Several formatting issues, e.g., ClebA in line 61, the CLIP in Figure 2.

**Ethical Concerns:**

["NO or VERY MINOR ethics concerns only"]

**Final Justification:**

The authors resolve my major concern about perceptual quality metrics and lack of adaptive attacks. While there still exist some small weaknesses, I recognize the authors' response and believe that they would be fixed well. Therefore, I recommend borderline accept.

**Limitations:**

Yes.

**Paper Formatting Concerns:**

No.

**Quality:**

3

**Strengths And Weaknesses:**

Strengths
1) New Problem: The paper clearly identifies an underexplored but increasingly relevant threat model: unauthorized style mimicry in T2I models.

2) Targeted Defense Design: The idea of style-specific imperceptible perturbations is elegant. The method is compatible with real-world deployment since it operates on the artist’s side, requires no model access, and preserves visual quality.

3) Empirical Results: Quantitative and qualitative results across multiple metrics demonstrate that the perturbations are effective in breaking style imitation.

Weaknesses

1) Overfitting Risk: The defense relies on a style classifier trained on few samples from each artist, which may limit generalization to unseen prompts or model variants. Additional experiments on style drift (e.g., over long prompts or different seeds) would improve robustness evaluation.

2) Lack of Standardized Attack Protocols: The paper evaluates mimicry via a style classifier and human judgments, but does not simulate realistic attacker pipelines. Incorporating these could make the evaluation more comprehensive.

3) Lack of Experiments: While many experiments are conducted, there is no experiments on user study and adaptive attacks.

---

> ### Author Rebuttal · Authors · 2025-07-31
>
> We sincerely appreciate the reviewer's valuable suggestions and constructive feedback. In this response, we address the reviewer's concerns point by point.
>
> **Question 1**: StyleGuard assumes a relatively narrow threat model where the attacker mimics an artist’s style using only prompt-based queries to a T2I model. However, it does not consider more advanced or adaptive adversaries, such as those employing prompt inversion techniques, personalized fine-tuning via DreamBooth or LoRA, or combining reference images with prompts for multimodal imitation.
>
> **Response:** **Our method explicitly considers prompt inversion (Table 4), DreamBooth (Table 1), and LoRA fine-tuning (Table 3)** in our main paper and appendix. Experimental results demonstrate superior performance over baselines. While previous defenses, such as Mist and AntiDreambooth, focused primarily on DreamBooth attacks, our style loss formulation inherently provides protection against various attack modalities simultaneously.
>
> **Question 2**: The approach heavily relies on a style classifier trained per artist to guide perturbation optimization.
>
> **Response:** Our method **does not need a style classifier trained per artist to guide perturbation optimization**. Our approach uses general pre-trained feature extractors (VGG/VAE) to derive style representations from intermediate layer statistics (mean and variance), eliminating the need for artist-specific classifiers. This ensures scalability without compromising protection effectiveness.
>
>
> **Question 3**: Lack of perceptual quality metrics
>
> We did not report perceptual quality metrics in our paper because we use the same attack budget (8/255) compared with the baseline methods. Because we cannot provide images for visualization, to further investigate the perceptual quality under different budgets, we conduct attacks under the same budget (8/255), and then compare the visual quality results with the baselines. We compared them with the original images using two widely adopted metrics: LPIPS [1] and SSIM [2]. LPIPS measures perceptual difference (lower = better quality). SSIM evaluates structural similarity (higher = better detail preservation)
>
> We compared against baseline approaches: Glaze, Mist, Anti-Dreambooth, SimAC, and MetaCloak.  Our method achieves **the best** LPIPS (`0.02`) and SSIM (`0.87`), which outperforms SimAC (LPIPS: `0.03`, SSIM: `0.82`) and MetaCloak (LPIPS: `0.03`, SSIM: `0.87`).
>
> ### Table 1: Visual Quality Comparison between Original and Perturbed Images.
> | Method          | LPIPS ↓ | SSIM ↑  |
> |-----------------|------------------|---------|
> | Original Images | 0.00             | 1.00    |
> | Glaze           | 0.06             | 0.80    |
> | Mist            | 0.07             | 0.85    |
> | Anti-Dreambooth | 0.15             | 0.85    |
> | SimAC           | **0.03**         | 0.82    |
> | MetaCloak       | 0.03             | 0.85    |
> | **Our Method**  | **0.02**         | **0.87**|
>
>
> **Question 4**: Need for human evaluation
>
> **Response:** We evaluated our method using multiple quantitative metrics, including FID, Precision, and CMMD, to rigorously assess its effectiveness. Additionally, **we conducted a user study (detailed in Figure 4 of our paper)** where participants were asked: "Based on the image style and quality, which image better fits the reference samples?" The results demonstrated that, compared to baseline methods, our approach achieved the lowest success rate in style mimicry, confirming its superior protection capability.
>
> **Question 5**: Potential interference with legitimate use
>
> **Response:** Artists can always use unprotected versions of their own work for artwork expansion. Our method specifically targets unauthorized mimicry attempts while preserving legitimate usage rights.
>
> **Question 6**: Since StyleGuard depends on a style classifier trained per artist to generate perturbations, how well does this classifier generalize across prompts, unseen styles, or different diffusion models? Is there a risk that the protection degrades under cross-model or cross-domain mimicry attempts?
>
> **Response:** As mentioned earlier, our method does not rely on a specific style feature extractor, but an ensemble of pretrained image encoders, such as VGG VAE and CLIP. This design ensured that the generalizable style features are extracted. Through the style loss, we make it closer to the target style and farther away from the original style. The results showed that our method is generalizable to different models (Table 2 in our paper). Moreover, our method is also generalizable to different styles.
> Table 2 shows the results of using images with different styles as the target images. Note that we **do not** need to train a specific style classifier/extractor but use a general style extractor, such as VAE and VGG, to extract the style-related features. It is shown that we can successfully defend against the attacks by using target images from different styles.
>
> #### Table 2: Evaluating the influence of the target styles
> | Style   | Sketch | Stick Figure | Post Impression | Ink Splash | Surrealism | Periodic Image | Gray Photo | Realism | Watercolor Painting | Cubism |
> |--------------------|--------|--------------|------------------|------------|------------|------------------|------------|---------|----------------------|--------|
> | **CMMD Distance to Training Dataset**  | 4.67  | 4.72| 4.85 | 5.15      | 5.71      | 6.13  | 6.42 | 6.81   | 6.88     | **7.58**  |
> | **FID**| 167.07 | 179.35 | 178.54  | 173.18  | 161.45    | 183.99 | 183.2     | **223.92** | 196.95    | 212.71 |
> | **CMMD**  | 1.83  | 2.20| 2.45  | 1.20| 1.60  | 2.86| 1.016 | 3.319   | 3.17      | **3.65**  |
> | **Precision**     | 0.20  | 0.37| 0.16 | 0.25 | 0.29 | **0.04**| 0.22| 0.09    | 0.12     | 0.08  |
>
> To evaluate the cross-domain mimicry attacks, we conduct experiments on the CelebA dataset, which contains faces from different identities. The results show that our method can successfully prevent identity customization attacks, as shown in Table 5 and Figure 7 in our paper.
>
> **Question 7**: The paper does not include experiments on adaptive attacks.
>
> **Response:** Thank you for inspiring further ideas for our experimental design. We admit that our present work did not consider the adaptive attacks because all of the current purifier-based attacks use pretrained diffusion models. We agree that an adaptive attacker that has defense knowledge poses more of a threat to previous methods and to ours. We will explore the feasibility of introducing adaptive attackers in experimental settings to test the limits and robustness of our method in future work.
>
> **Question 8**: There is no accompanying description or discussion for Table 2, which makes it difficult to interpret its relevance and implications. Furthermore, the reference in Line 283 appears to mistakenly point to Table 3 instead.
>
> **Response**: A comprehensive analysis of Table 2 has been provided in lines 283–288 of our paper. The citation in line 283 correctly refers to Table 2 (verified upon careful re-examination).
> Line 294 corresponds to the reference for Table 3. Tables 2 and 3 are on the same row, which may be the reason for misleading the reviewer.
>
> **Response to Q9:**
> Thanks for your suggestion. We will revise these in our final version.
>
> We hope our explanations address the reviewers' questions. We welcome any further questions.

---

> > ### Comment · Reviewer_rV8Y · 2025-08-02
> >
> > I thank the authors for their clarification. I think this paper requires further revision based on the current weaknesses. Furthermore, I still have the following concerns.
> > 1. If StyleGuard relies on an ensemble of encoders, could you clarify how you ensure that the extracted features reliably capture style-specific information without task-specific finetuning? Additionally, how do you prevent from overfitting of these particular encoders?
> > 2. I believe adaptive attacks are necessary in this paper because they represent a more realistic and challenging threat model. Without evaluating under adaptive settings, it is difficult to assess whether the proposed defense is truly robust or simply effective under certain attack scenarios. This is especially important given the paper’s claim of generalizability and model-agnostic protection.
> > 3. The explanation to tables in the text, as well as their captions, reduces the readability of the paper. I suggest authors carefully revise the paper.

---

> > > ### Author Response · Authors · 2025-08-02
> > >
> > > [1] As discussed in our paper, we utilize the mean and variance of the features to represent the style characteristics of an image. This approach has been widely validated in previous studies for effective style translation, notably in works such as [1] and [2].
> > >
> > > [2] We have thoroughly evaluated our method against the latest attacks, including DiffPure and Noise Upscale. The latter was recently presented at ICLR 2025. To the best of our knowledge, our work is the first to successfully defend against these attacks. Moreover, no adaptive attack targeting mimicry attacks has yet been proposed. The use of pretrained models, such as VAE and VGG, inherently mitigates overfitting risks due to their training on millions of diverse images.
> > >
> > > [3] As clarified in our response, all tables in our manuscript have been accurately illustrated and properly referenced.
> > >
> > > We appreciate the reviewer’s comments, although there were many basic mistakes in the initial review. No one wants to be a victim of AI review. We hope this response facilitates a more comprehensive and accurate understanding of our contributions.
> > >
> > > [1] Encoding in Style: a StyleGAN Encoder for Image-to-Image Translation, CVPR 2021
> > > [2] Arbitrary Style Transfer in Real-time with Adaptive Instance Normalization, 2017

---

> > > > ### Comment · Reviewer_rV8Y · 2025-08-02
> > > >
> > > > Thanks for the response. I believe the authors may have misunderstood the intent behind the term adaptive attack. It does not refer to whether your defense can withstand existing attacks like DiffPure or Noise Upscale. Success against some existing attacks indicates that the defense method may address some known vulnerabilities in prior attacks. However, an adaptive attack refers to a threat model where the attacker has partial knowledge of your defense's design principles, assumptions, or objectives, and then tailors the attack accordingly to bypass it.
> > > >
> > > > Introducing a new defense naturally invites questions about its limitations. No defense can be universally robust against all future attacks. Therefore, it is important to analyze and discuss under what conditions your method might still be vulnerable, particularly in settings where the attacker adapts to your defense mechanisms. This kind of analysis is essential for understanding the true robustness and generalizability of your approach.

---

> ### Author Response · Authors · 2025-08-03
>
> We propose a simple adaptive attack method in which the attacker is aware of the purifier model used to generate the perturbation but employs a different purifier model to remove the noise. The results are summarized in the table below. The terms X2 and X4 refer to different stable-diffusion-based upscale models used to compute the upscale loss, while SinSR denotes the purifier employed by the attacker. Our findings indicate that utilizing a purifier with a different architecture can reduce the effectiveness of the protection to some extent. However, by leveraging an ensemble of purifiers when calculating the upscale loss, the overall protection performance is significantly enhanced.
>
> | | X2    |  || X4  |  |  | X4+X2     |  |  |
> |-------|-----------|-----------|-----------|-----------|-----------|-----------|-----------|-----------|-----------|
> |  | **FID** | **CMMD**| **PRE**   | **FID**   | **CMMD**  | **PRE**   | **FID**   | **CMMD**  | **PRE**   |
> | **X2**    | 197.637   | 2.9828    | 0.208     | 186.303   | 0.56      | 0.341     | 204.979   | 2.879     | 0.166     |
> | **X4**    | 188.308   | 2.205     | 0.25 | 191.35    | 2.748     | 0.2916    | 253.939   | 2.735     | 0.166 |
> | **SinSR** | 167.489   | 0.64  | 0.458 | 183.119   | 1.116     | 0.208     | 183.643   | 1.887 | 0.163 |

---

> ### Author Response · Authors · 2025-08-05
> **Supplementary Experiments on Q7 (Adaptive Attacks)**
>
> To further investigate the threat of adaptive attacks, we design a more advanced adaptive attack where the adversary has access to both clean and perturbed images. Using these paired images, the adversary fine-tunes a purifier model to effectively remove protective perturbations. Specifically, we first employ X2 and X4 as shadow purifiers to generate protective perturbations on 200 images. Then, following the fine-tuning method from SinSR, we fine-tune a ResShift [1] purifier. After training, the purifier is used during the test phase to remove the protective noise, and subsequently, DreamBooth is applied to fine-tune a Stable Diffusion model. As shown in the table below, experimental results demonstrate that the fine-tuned ResShift purifier significantly degrades the effectiveness of the protection, highlighting the practical threat posed by adaptive attacks. However, such an adaptive attack relies on the attacker being able to obtain a large number of clean and perturbed corresponding samples. In the case that the attacker cannot obtain a large number of paired samples, our attack still has a strong protective effect.
>
> | Model    | X2+X4 |  |  |
> -----------------|---------|--------|-------|
> |        |FID     | CMMD   | PRE   |
> | X2              | 204.979 | 2.879  | 0.166 |
> | X4              | 253.939 | 2.735  | 0.166 |
> | SinSR           | 183.643 | 1.887  | 0.163 |
> | Finetuned ResShift | **164.562** | **0.793**  | **0.622** |
>
>
> [1] ResShift: Efficient Diffusion Model for Image Super-resolution by Residual Shifting, NIPS 2023

---

> > ### Comment · Reviewer_rV8Y · 2025-08-06
> >
> > Thanks for your response. The proposed adaptive attack setup still has important limitations. It assumes the attacker has access to a large number of clean and perturbed image pairs and can fine-tune a purifier model accordingly. This is a strong assumption and does not cover more realistic adaptive scenarios where the attacker only knows the defense design (e.g., the use of an ensemble of purifiers or the type of loss used) but does not have access to paired data.
> >
> > Moreover, the current adaptive attack only evaluates a narrow form of adaptation, i.e., retraining a purifier. It does not explore alternative attack strategies that directly optimize against the defense objective or exploit potential weaknesses in the purifier ensemble. As a result, the evaluation remains incomplete, and it is still unclear whether the defense is truly robust under a broader range of adaptive threats.
> >
> > I also believe the current treatment of adaptive attacks remains insufficient. When a paper's core contribution is to propose a new defense, a thorough investigation and discussion of adaptive attacks is not optional but necessary. This aspect should occupy a meaningful portion of the paper, ideally half a page to a full page, not just be an afterthought during the rebuttal. I also noted that other reviewers have raised similar concerns regarding adaptive attacks. I strongly encourage the authors to reflect more deeply on this issue, rather than rushing to add experiments without fully considering the implications.
> >
> > Given other weaknesses in the original submission, I would like to maintain my current score and recommend a major revision.

---

> ### Author Response · Authors · 2025-08-06
>
> According to the review guideline， reviewers are subject to Responsible Reviewing initiative, including the desk rejection of his/her co-authored papers for grossly irresponsible behaviors. We hope the reviewer rV8Y could read our paper carefully instead using AI to generate content. Our paper are acknowledged by the other two reviewers.

---

> ### Comment · Reviewer_rV8Y · 2025-08-06
>
> Hi, authors. I would like to clarify that I have carefully read your paper and taken time to understand its design and contributions. If I have misunderstood any part of your work, I sincerely welcome clarification and further discussion. I believe some of our disagreement stems from differing interpretations of what constitutes an appropriate adaptive attack in your setting, so let me explain in more detail.
>
> From my reading, the core of your defense lies in designing a new style loss and leveraging a combination of purifiers and upscalers to generate perturbations. Therefore, an adaptive attacker, in my view, should be modeled based on knowledge of this design. In Section 4.1, the paper mentions that an adversary can access images and finetune a text-to-image generator. I assume your adaptive attack is based on a similar assumption, where the attacker finetunes a purifier using original images and perturbed images to weaken the protection.
>
> Where I disagree is the assumption that an attacker would have access to a large set of perturbed images generated by your defense. If the attacker only knows the defense structure (the use of a style loss or ensemble of purifiers) but does not have access to the actual perturbed outputs, how would they obtain the data needed to train a purifier?
>
> Additionally, I find the attack approach of training a purifier itself somewhat confusing, as it does not appear to directly exploit the core weakness of the defense. In contrast, I believe a more meaningful adaptive attack should test whether, given core knowledge of the defense (e.g., the design of style loss), the attacker can manipulate the input image (via adding perturbation or somehow) in such a way that the defense system still outputs an unprotected image.
>
> If I have misunderstood your design choices, I would be glad to hear your perspective.

---

> ### Author Response · Authors · 2025-08-07
>
> Thanks for your further comments. We will clarify the concerns. We have designed two different adaptive attacks. In the simple adaptive attacks, the attacker only knows the ensemble purifiers used for training and uses a different upscaler in the test phase.
> In this case, the attacker uses a different upscaler, as shown in the table below. The terms X2 and X4 refer to different stable-diffusion-based upscale models used to compute the upscale loss, while SinSR [1] denotes the purifier employed by the attacker. Our findings indicate that utilizing a purifier with a different architecture can reduce the effectiveness of the protection to some extent. However, by leveraging an ensemble of purifiers when calculating the upscale loss, the overall protection performance is significantly enhanced.
>
> | | X2    |  || X4  |  |  | X4+X2     |  |  |
> |-------|-----------|-----------|-----------|-----------|-----------|-----------|-----------|-----------|-----------|
> |  | **FID** | **CMMD**| **PRE**   | **FID**   | **CMMD**  | **PRE**   | **FID**   | **CMMD**  | **PRE**   |
> | **X2**    | 197.637   | 2.9828    | 0.208     | 186.303   | 0.56      | 0.341     | 204.979   | 2.879     | 0.166     |
> | **X4**    | 188.308   | 2.205     | 0.25 | 191.35    | 2.748     | 0.2916    | 253.939   | 2.735     | 0.166 |
> | **SinSR** | 167.489   | 0.64  | 0.458 | 183.119   | 1.116     | 0.208     | 183.643   | 1.887 | 0.163 |
>
> For the style loss, we propose an adaptive attack scenario in which the attacker is assumed to have knowledge of the style encoder and the target image. The attacker optimizes the input image’s features to closely match those of the original image while simultaneously distancing them from the target image’s features. We conducted experiments under this scenario, where no purifier was applied during testing; instead, the image was re-optimized by maximizing the style loss. Results show that this adaptive attack reduces the effectiveness of our protection. However, due to the presence of the denoise loss and upscale loss components, our method still provides a certain level of defense.
> | |FID|CMMD|PRE|
> |-------|-----------|-----------|-----------|
> |Clean Imgae| 142.38|0.65|0.72|
> |Noised Image|197.63|2.98|0.20|
> |Reoptimized Image|182.51|1.89|0.28|
>
> We thank the reviewer for suggesting the possibility that the attacker pre-perturbs the image. We should note that in our attack scenario, the images are first posted online by the artist and then downloaded by users. A malicious user could download these images, remove the protective noise, and then perform the attack. Therefore, we did not consider the possibility that the attacker pre-perturbs the image.
>
> In the strong adaptive attacks, we suppose the attacker can download hundreds of both clean and perturbed images online. We admit that this is a strong assumption, but it also shows that our defense will not be easily breached.
>
> [1]  SinSR: diffusion-based image super-resolution in a single step, CVPR2024

---

> > ### Comment · Reviewer_rV8Y · 2025-08-07
> >
> > Well, I think the authors have well understood what I am concerned about. I am happy to see that the adaptive attack section looks complete now. Since the authors have resolved one of my main question that I was originally inclined to reject, I think there is no reason not to raise my score.
> >
> > I hope the authors could allocate sufficient space for discussing adaptive attacks, and make sure the questions raised by all the reviewers would be well-resolved and reflected in the final verision if this paper is accepted. I can imagine a lot of work needs to be done. Good luck!

---

> > > ### Author Response · Authors · 2025-08-07
> > >
> > > We would like to sincerely thank the reviewers for their thorough review and constructive suggestions, which greatly improved the quality of this paper.

---

### Official Review · Reviewer_bWmD · 2025-07-03

**Clarity:** 3
**Significance:** 2
**Originality:** 2
**Rating:** 4
**Confidence:** 3

**Summary:**

This paper proposed StyleGuard as an end-to-end defence against style mimicry attacks. The defensive perturbation is produced by alternatively updating the diffusion model and the PGD updates on images. There are three terms to drive the udpate: denoise loss, style loss, and upscale loss. The authors evaluated StyleGuard on WikiArt and CelebA to demonstrate that the method raises FID and drops CLIP precision even after DiffPure/Noise-Upscale purification.

**Questions:**

See above.

**Ethical Concerns:**

["NO or VERY MINOR ethics concerns only"]

**Final Justification:**

- The authors did ablation studies regarding $\lambda$, $\eta$, $K_2$ and $K_2$ thouroughly and provided the cost compared to the baseline.
- Regarding the question of adaptive attack, the authors did not give a specific solution but said they will explore in future work.
- The authors further discussed the defense bound with ablation on different attack settings (X2, X4, SinSR).
Overall, I feel the authors solved my concerns, regarding the adaptive attack, I feel it might be good to mark this in the limitation and encourage future exploration. Therefore I would recommend score 4.

**Limitations:**

yes.

**Paper Formatting Concerns:**

NA.

**Quality:**

2

**Strengths And Weaknesses:**

- **Method**. The design of the loss is startghtforward but deals with multiple intents. The upscale loss enables StyleGuard to anticipate purifiers' erasing and it uses an ensemble of purifiers as noise that beats DiffPure might be dropped by an upscaler. This upscale loss serves as StyleGuard's insurance to make the defensive perturbation survive standard clean up mechanisms. So such design makes sense. Another ingredient style loss perturbs global latent statistics, which is the weakness of Mist, to make StyleGuard robust to cropping/rescaling/Gaussian noise.

- **Empirical**. The evaluation is done on DreamBooth and Textual-inversion, full/LoRA fintuning, and black-box upscalar FIE. And the model is acorss SDV1.4/1.5/2.1/XL. I think the evaluation part is thorough.

Limitations \& Questions:

(1) $\lambda$, $\eta$, budget, K1, K2 parameters are fixed, Have the authors done ablation study to see how sensitive is the method related to these parameters?

(2) Can the authors provide detailed computational cost of each phase, with comparison to baselines?

(3) The purification is emprical without certified bound. DiffPure is frozen in the upscale loss phase. consider an adaptive attacker that can jointly finetune the purifiers with DreamBooth such that the gradient can realign. Have the authors consider adpative attacks? Or consider how StyleGuard can provide/further develop certified bound? An ablation with different level of bugdet discussion might be helpful.

(4) Minor. It is better to not use the abbreviation PGD in line 34 (first use in the paper)

Overall, the idea is straightforward to follow and its underlying idea remains in line with established methodologies although achieves good empirical results. If the authors can resolve my above concerns, I am willing to raise my score.

---

> ### Author Rebuttal · Authors · 2025-07-28
>
> We sincerely thank the reviewers for the constructive comments. In response to Q1, we add ablation studies and detailed explanations of the metrics. In response to Q2, we compare the time cost of our method with the baseline method.
>
> ### Response to Q1
> We conducted ablation studies on the style loss weight $\lambda$, the attack budget ($B_{\inf}$ in Eq.8), upscale loss weight $\eta$, $K_1$, and $K_2$ to explore their influence on the attack effect. To assess the generated image qualities of the fine-tuned SD model, we adopt two additional metrics, in addition to FID and Precision, including LPIPS[1] and CMMD[2].  CMMD is an improvement metric for FID, which can better measure the image distortions. The higher the value, the greater the gap with the training dataset.
>
> We first generate perturbations on the SD1.4 and Upscaler x4. The training dataset includes 15 images from Van Gogh. Then we upscale the images using a different upscaler x2, and fine-tune the SD1.4 and SD1.5, respectively, using DreamBooth. In all experiments, the first stage (generating protection noise) trains for 10 epochs. The second stage (Dreambooth fine-tuning) trains for 200 epochs. We also fine-tune an SD1.4 and an SD1.5 with clean images as the references.  Next, we use the reference models and adversarially fine-tuned models to generate the reference image set $I_{ref}$ and the attacked image set $I_{attack}$ under the same prompt (such as "an \<sks\> painting of a house). Finally, we compute the above metrics using  $I_{ref}$ and $I_{attack}$. The results are shown in Table 1 and Table 2.
> #### Table 1: Ablation Study on the Style Loss Weight (Dreambooth Finetuning on SD14)
> | $\lambda$ | FID| Precision | Recall  | LPIPS  | CMMD   |
> |-------|-------|-----------|---------|--------|--------|
> | 1 | 197.38| 0.37 | 0.63   | 0.70    | 3.05  |
> | 10| $\mathbf{214.27}$ | $\mathbf{0.25}$      | 0.63   | $\mathbf{0.71}$   | $\mathbf{4.02}$  |
> | $10^2$ | 194.84| 0.41| 0.58    | 0.69  | 3.09  |
> | $10^3$ | 188.64| 0.52| $\mathbf{0.33}$    | 0.65  | 2.22  |
>
> #### Table 2: Ablation Study on the Style Loss Weight (Dreambooth Finetuning on SD15)
> | $\lambda$  | FID   | Precision | Recall  | LPIPS  | CMMD   |
> |-------|-------|-----|----|--------|--------|
> | 1 | 184.32| 0.29| 0.83   | 0.68  | 2.37|
> | 10| $\mathbf{227.20}$| $\mathbf{0.25}$| 0.72| $\mathbf{0.70}$  | $\mathbf{3.22}$   |
> | $10^2$| 194.32| 0.41| 0.50| 0.67  | 2.18  |
> | $10^3$| 180.21| 0.54| $\mathbf{0.41}$| 0.63| 1.69  |
>
> Experiments show that as $\lambda$ increases, FID, LPIPS, and CMMD first increase and then decrease, and the Precision first decreases and then increases. This indicates that there is a balance between denoising loss and style loss. This could be attributed to the fact that the two different targets of the denoising loss and style loss may not always be consistent and might be partially contradictory to each other. The aim of the denoising loss is to pull the representation of the image out of the feature space of the diffusion model, and the aim of the style loss is to make the style features of the generated image closer to the target image and further from the original image.
>
> Table 3 displays an ablation study on the upscale loss weight, $\eta$. The findings indicate that a low weight of 0.1 results in the least effective protection, evidenced by a low FID score of 135.466 and a precision of 0.625. In contrast, higher weights, such as 10, offer better protection. However, as the value of $\eta$ increases from 10 to 100, the FID and the CMMD decrease. This trade-off implies the existence of a balanced weight that maximizes performance across the metrics.
> #### Table 3:  Ablation Study on the Upscale Loss Weight
> |$\eta$|FID|Precision|LPIPS|CMMD|
> |-----------|---|---------|-----|-----|
> |0.1|135.466|0.625|0.625|0.822|
> |1|173.399|0.375|$\textbf{0.686}$|1.366|
> |10|$\textbf{184.747}$|$\textbf{0.167}$|0.677|$\textbf{1.711}$|
> |100|182.477|0.333|0.665|0.644|
>
> The ablation study of the attack budget is shown in Table 4.  It is shown that as the budget increases, FID and CMMD gradually increase, while Precision gradually decreases. When the budget is higher than 0.063, increasing the budget does not significantly improve the protection effect. This is because the difference in the generated images is already large enough. In addition, a too high budget will introduce visible protection noise, which will damage the quality of the protected image.
>
> #### Table 4: Ablation Study on the Attack Budget (Dreambooth Finetuning on SD15)
> | Budget | FID | Precision | Recall  | LPIPS  | CMMD   |
> |--------------|-------|--------|------|-----|-----|
> | 2/255| 165.76 | 0.39 | 0.52 | 0.60  | 1.09 |
> | 4/255| 199.78 | 0.29 | 0.77| 0.62  | 1.15 |
> | 8/255| 190.99 |  0.24 | $\textbf{0.50}$   | 0.63  | 2.61  |
> | 16/255| $\textbf{271.80}$ |$\textbf{0.04}$ |0.87| 0.67 |3.55 |
> | 32/255| 256.23 | 0.08 | 0.81| $\textbf{0.70}$  | $\textbf{3.71}$  |
>
> Table 5 shows the ablation study on the $K_1$ and $K_2$. We highlight the top two best values for each metric in bold.  It can be observed that when $K_1$=10 and $K_2$=5, better protection effects are achieved across four metrics. The next best results are achieved when $K_1$=5, $K_2$=5, with good protection effects across three metrics. The results also show that the protection effect is significantly weakened when the difference between K1 and K2 is large (for example, when K1=5, K2=20 or K1=20, K2=5, the CMMD values are only 0.674 and 0.704). We think this may be because when the difference between K1 and K2 is large, the noise is more likely to fall into overfitting.
> #### Table5: Ablation Study on $K_1$ and $K_2$ (Dreambooth Finetuning on SD15)
> |$K_1$|$K_2$|FID|Precision|Recall|LPIPS|CMMD|
> |---|---|---|---|---|---|---|
> |5|5|**211.728**|0.292|0.169|**0.682**|**2.314**|
> |5|10|155.110|0.291|0.472|0.665|1.369|
> |5|20|146.956|0.458|0.444|0.676|0.674|
> |10|5|**231.200**|**0.167**|0.889|**0.705**|**2.472**|
> |10|10|164.830|0.25|0.667|0.671|1.725|
> |10|20|144.150|0.625|**0.167**|0.658|0.646|
> |20|5|163.648|0.208|0.194|0.649|0.704|
> |20|10|194.953|**0.125**|**0.168**|0.660|1.734|
> |20|20|180.729|0.168|0.583|0.677|2.245|
>
> ### Response to Q2
> Table 6 details the computational time costs associated with different loss functions, while Table 7 provides a comparison between our approach and the baseline method, SimAC. Both methods were evaluated on two RTX 3090 GPUs with a batch size of 4, using parameters K1=10 and K2=5. The tables present the average time cost for various stages, the total time per epoch, and the total attack duration for each method.  Our method incurs a 36% higher time cost per epoch compared to SimAC. This is due to the need to compute three different losses during the PGD attack stage, whereas SimAC computes only two losses (feature loss and denoising loss). Despite the increased time per epoch, our method achieves a ~50% reduction in overall attack time compared to SimAC. This advantage arises because SimAC requires an additional 10 minutes for diffusion step selection before initiating the attack, which our method does not require.
>
> #### Table 6: Time Cost for Different Losses
> |Loss |Style Loss|Denoising Loss|Upscale Loss|
> |-----|----------|------|-------|
> |Time cost | 0.11s |0.12s|0.14s|
>
> #### Table 7: Comparing the Time Cost with the Baseline
> | Method | Pretraining | PGD attack | Post-training | Average time per epoch (batch size=4) | Total time (10 epochs) |
> |-----|-----|---------|-------|------|---------|
> | SimAC  | 3.14s | 2.98s | 2.72s  | 36.25s  | 16.25 min|
> | Ours   | 3.42s | 5.84s| 2.98s  | 49.32s | 8.22min|
>
> ### Response to Q3
> Thank you for inspiring further ideas for our experimental design. This is a very important suggestion. While current purifier-based attacks mainly focus on using pretrained diffusion models because of the high computation cost of training a new diffusion-based purifier, we agree that an adaptive attacker that can fine-tune a purifier on protected images poses more of a threat to previous methods and to ours. We will explore the feasibility of introducing adaptive attackers in experimental settings to test the limits and robustness of our method in future work.
>
> To explore the defense bound, we use a novel purifier, SinSR, in the attack phase, which is unseen in the training. The result is shown in Table 5 in the response to Reviewer 1. We compute the upscale loss on three different settings (X2, X4, and X2+X4), and then use three different purifiers in the attack phase, including X2, X4, and SinSR [4]. The result shows that SinSR is able to achieve a certain attack effect. However, when both X2 and X4 are used for the upscale loss, the protection performance on the SinSR slightly improved. This shows that using an ensemble of purifiers can improve the protection strength. We also conducted the ablation study on the budget and show the results in Table 4.
>
> Table 5: Ablation Study on the Attack Budget (Dreambooth Finetuning on SD15)
> | | X2  |  || X4  |  |  | X4+X2     |  |  |
> |-------|--------|---------|-----------|-----------|-----------|-------|-----------|-----------|-----------|
> |  | FID | CMMD| PRE| FID | CMMD| PRE| FID | CMMD| PRE|
> | **X2**    | 197.637| 2.9828| 0.208| 186.303   | 0.56  | 0.341 | 204.979   | 2.879     | 0.166     |
> | **X4**    | 188.308| 2.205| 0.25 | 191.35| 2.748 | 0.2916 | 253.939   | 2.735     | 0.166 |
> | **SinSR** | 167.489| 0.64| 0.458 | 183.119| 1.116| 0.208  | 183.643   | 1.887 | 0.163 |
>
> Finally, thank you again for your in-depth analysis and valuable suggestions regarding our work.
>
> Reference
> [1]  The unreasonable effectiveness of deep features as a perceptual metric. CVPR, 2018.
> [2]  Rethinking FID: Towards a Better Evaluation Metric for Image Generation. Arkiv2401.09603, 2024
> [3]  Improved Precision and Recall Metric for Assessing Generative Models. NIPS, 2019
> [4] SinSR: diffusion-based image super-resolution in a single step, CVPR2024

---

> ### Author Response · Authors · 2025-08-04
>
> Dear Reviewer bWmD,
>
> We hope this message finds you well.
>
> We have done all the experiments according to your review. As the rebuttal period deadline is approaching, we would greatly appreciate it if you could kindly respond to our rebuttal. We are looking forward to receiving any further questions or suggestions for improvement. Your support is important to our work. We thank you sincerely for reading our paper carefully, especially considering that the reviewer rV8Y used AI to generate all the comments, and most do not reflect the facts in the paper.
>
> Best regards,
>
> Authors of NeurIPS 13927

---

> ### Author Response · Authors · 2025-08-05
> **Supplementary Experiments on Q3 (Adaptive Attack)**
>
> To further investigate the threat of adaptive attacks, we design a more advanced adaptive attack where the adversary has access to both clean and perturbed images. Using these paired images, the adversary fine-tunes a purifier model to effectively remove protective perturbations. Specifically, we first employ X2 and X4 as shadow purifiers to generate protective perturbations on 200 images. Then, following the fine-tuning method from SinSR, we fine-tune a ResShift [1] purifier. After training, the purifier is used during the test phase to remove the protective noise, and subsequently, DreamBooth is applied to fine-tune a Stable Diffusion model. As shown in the table below, experimental results demonstrate that the fine-tuned ResShift purifier significantly degrades the effectiveness of the protection, highlighting the practical threat posed by adaptive attacks. However, such an adaptive attack relies on the attacker being able to obtain a large number of clean and perturbed corresponding samples. In the case that the attacker cannot obtain a large number of paired samples, our attack still has a strong protective effect.
>
> | Model    | X2+X4 |  |  |
> -----------------|---------|--------|-------|
> |        |FID     | CMMD   | PRE   |
> | X2              | 204.979 | 2.879  | 0.166 |
> | X4              | 253.939 | 2.735  | 0.166 |
> | SinSR           | 183.643 | 1.887  | 0.163 |
> | Finetuned ResShift | **164.562** | **0.793**  | **0.622** |
>
>
> [1] ResShift: Efficient Diffusion Model for Image Super-resolution by Residual Shifting, NIPS 2023

---

> > ### Comment · Reviewer_bWmD · 2025-08-05
> >
> > Thanks the authors for the detailed rebuttal. I will raise my score from 3 to 4 in my final justification.

---

> > > ### Author Response · Authors · 2025-08-05
> > >
> > > We sincerely thank the reviewer, bWmD, for the detailed review of our paper, as well as the recognition and recommendation!

---

### Official Review · Reviewer_YoSP · 2025-07-06

**Clarity:** 3
**Significance:** 3
**Originality:** 4
**Rating:** 4
**Confidence:** 4

**Summary:**

This paper introduces StyleGuard, a novel method to protect artists' work from style mimicry attacks by text-to-image diffusion models like Stable Diffusion. The core problem it addresses is that existing protection methods (e.g., Glaze, Anti-DreamBooth) are vulnerable to new "purification" attacks (e.g., DiffPure, Noise Upscaling), which can remove the protective noise. Furthermore, existing defenses often lack transferability, failing to protect against unknown or black-box models.

StyleGuard's main contributions are two novel loss functions used to generate a robust, imperceptible perturbation on images:
1.  A "Style Loss" that manipulates the mean and variance of features in the latent space. This perturbs the global style characteristics of an image, making it difficult for models to learn the correct style and enhancing the protection's transferability across different text-to-image models.
2.  An "Upscale Loss" designed to specifically counter purification attacks. It trains the perturbation against an ensemble of different purifiers and upscalers, aiming to maximize the denoising error and essentially making the protective noise "purification-resistant."

The authors validate StyleGuard through extensive experiments on the WikiArt and CelebA datasets. The results show that their method significantly outperforms existing baselines in robustness against various data transformations and, crucially, against state-of-the-art purification methods.

**Questions:**

Thank you for this interesting work. The proposed ideas are novel, but I have concerns about the methodological rigor and the breadth of the evaluation, which prevent me from giving a higher score at this time. My questions are as follows, ordered by importance:

1. The `Style Loss` depends on a target image from a different genre, a critical step described as being "randomly selected." How sensitive is the protection efficacy to the choice of this target? For instance, does a target style that is maximally dissimilar provide stronger protection, and is there a systematic way to choose an optimal target?

2. The methodology relies on several key hyperparameters, such as the `Style Loss` weight (λ=10) and the specific ensemble of purifiers for the `Upscale Loss`. Could you provide ablation studies that analyze the sensitivity to these parameters and offer a justification for the chosen values?

3. The evaluation focuses on individual purification methods. To better assess real-world robustness, have you considered testing against a pipeline of combined, simpler transformations (e.g., cropping, rotation)?

My final recommendation is contingent on the authors' response to these concerns. A more rigorous analysis of the method's sensitivity to its core parameters (Questions 1 & 2) and a more comprehensive robustness evaluation (Question 3) would be necessary to move this paper into a clear accept category.

**Ethical Concerns:**

["NO or VERY MINOR ethics concerns only"]

**Limitations:**

The authors provide a thorough analysis of the potential limitations of their method.

**Paper Formatting Concerns:**

I did not notice any major formatting issues.

**Quality:**

3

**Strengths And Weaknesses:**

This paper addresses a critical and timely problem with a novel solution. However, while the strengths in originality and significance are clear, they are counterbalanced by notable weaknesses in methodological rigor and the breadth of experimental validation, placing the paper on the borderline for acceptance.

Strengths:

1. The problem of efficiently protecting artists from AI-driven style mimicry is of great practical importance. The proposed StyleGuard framework, particularly its "Upscale Loss" designed to combat purification attacks, is a novel and conceptually clever contribution to this domain.

2. The experimental evaluation is mostly thorough. The authors compare StyleGuard against a comprehensive set of baselines and test against a strong and relevant suite of attacks, including the specific purification methods that this work aims to defeat.

3. The paper is generally well-written and easy to follow, with figures that help to illustrate the overall system architecture.

Weaknesses

1.  The core `Style Loss` requires a target image from a different art genre, but the selection process is vaguely described as "randomly selected." The choice of target style is critical, yet the paper does not explore how the degree of stylistic difference impacts protection efficacy, nor does it propose a systematic selection strategy. This ambiguity limits the method's reproducibility and reliability.

2. The algorithm relies on fixed hyperparameters, such as the ensemble of purifiers for the `Upscale Loss` and the style loss weight (λ=10). The paper lacks ablation studies to analyze the sensitivity to these parameters or to justify why these specific values were chosen. We cannot know if these choices are optimal or how robust the method is to their variation.

3.  The paper's evaluation of robustness is narrowly focused, primarily validating the defense against sophisticated, diffusion-based purifiers. However, it neglects to test against a wider range of transformations common in real-world scenarios. For instance, the paper does not evaluate robustness against a "stress test" of combined transformations (e.g., cropping, followed by rotation and compression). This limited scope makes it difficult to fully assess the protection's practical robustness.

---

> ### Author Rebuttal · Authors · 2025-07-27
>
> We sincerely thank the reviewer for insightful comments.  Below, we respond to the key points raised.
>
> ### Response to Q1
>
> To assess the impact of different target styles on our method, we selected ten distinct art genres, each represented by 30 images. We conducted simplified attacks while omitting the upscale loss to focus on style influence. To measure style variation, we utilized CMMD [1], an enhanced metric over FID, where a higher CMMD value indicates a greater divergence from the training dataset. The second row in the results reflects the CMMD distance of target style images compared to the training dataset.  The third to the fifth rows are the FID, CMMD, and Precision scores evaluated on the generated images. Here, we computed the style loss using the average mean and variance of style features on images in the same genre. **We find that a larger CMMD distance between the target style dataset and the training dataset correlates with a significant decline in generated image quality**, as evidenced by higher FID and CMMD values and lower Precision scores.
>
> #### Table 1: Evaluating the influence of the target styles
> | Style   | Sketch | Stick Figure | Post Impression | Ink Splash | Surrealism | Periodic Image | Gray Photo | Realism | Watercolor Painting | Cubism |
> |--------------------|--------|--------------|----------|------------|------------|------------------|------------|---------|------------|--------|
> | CMMD to Training Dataset | 4.67  | 4.72| 4.85 | 5.15 | 5.71| 6.13  | 6.42 | 6.81   | 6.88     | **7.58**  |
> | FID| 167.07 | 179.35 | 178.54  | 173.18  | 161.45    | 183.99 | 183.2  | **223.92** | 196.95    | 212.71 |
> | CMMD| 1.83  | 2.20| 2.45  | 1.20| 1.60  | 2.86| 1.016 | 3.319   | 3.17  | **3.65**  |
> | Precision| 0.20  | 0.37| 0.16 | 0.25 | 0.29 | **0.04**| 0.22| 0.09    | 0.12 | 0.08  |
>
> Inspired by this, we introduce a straightforward method for selecting the target image. This method evaluates their effectiveness using CMMD distances and FID scores. The complete process can be executed in under an hour utilizing 8 3090 GPUs.
>
> - Step 1: Begin by selecting 10 distinct art genres from an open-source dataset such as WikiArt.
> - Step 2: Calculate the CMMD distance between each image in these styles and the training dataset.
> - Step 3: Identify and select the image with the highest CMMD distance in each style.
> - Step 4: Perform a Simplified StyleGuard Attack
>   - For each of the 10 selected images:
>     - Train for 10 epochs using the selected target images while omitting the upscale loss.
>     - Using the generated images with protective perturbations to fine-tune the SD model for 200 epochs using DreamBooth.
>     - Record both FID (Fréchet Inception Distance), Precision, and CMMD values at the end.
> - Step 5: Choose the target image with the largest CMMD with FID>200 and Precision < 0.2 as the final target image for further optimization.
>
> ### Response to Q2
> Thanks for your valuable suggestions. We conducted ablation studies on the style loss weight $\lambda$, the attack budget ($B_{\inf}$ in Eq.8), upscale loss weight $\eta$, $K_1$, and $K_2$ to explore their influence on the attack effect. To assess the generated image qualities of the fine-tuned SD model, we adopt two additional metrics except FID and Precision, including LPIPS[1], and CMMD[2].
>
> We first generate perturbations on the SD1.4 and upscaler x4. Then we upscale the images using a different upscaler x2, and fine-tune the SD1.4 and SD1.5, respectively, using DreamBooth. In all experiments, the first stage (generating protection noise) trains for 10 epochs. The second stage (Dreambooth fine-tuning) trains for 200 epochs. We also fine-tune an SD1.4 and an SD1.5 model with clean paintings as the reference models.  Next, we generate the reference image set $I_{ref}$ and the attacked image set $I_{attack}$ under the same prompt. Finally, we compute the above metrics using $I_{ref}$ and $I_{attack}$. The results are shown in Table 2 and Table 3.
>
> Experiments show that as $\lambda$ increases, FID, LPIPS, and CMMD first increase and then decrease, and the Precision first decreases and then increases. This indicates that there is a balance between denosing loss and style loss. This could be attributed to the fact that the two different targets of the denosing loss and style loss may not be always consistent and might partially controdictary with each other, in which the aim of the denosing loss is to pull the representation of the image out of the feature space of the diffusion model, and the aim of style loss is to make the style features of the generated image close to the target image and away from the original image.
>
> #### Table 2: Ablation Study on the Style Loss Weight (Dreambooth Finetuning on SD14)
> | $\lambda$ | FID   | Precision | Recall  | LPIPS  | CMMD   |
> |--------------|-------|-----------|---------|--------|--------|
> | 1 | 197.38| 0.37     | 0.63   | 0.70    | 3.05  |
> | 10| $\mathbf{214.27}$ | $\mathbf{0.25}$      | 0.63   | $\mathbf{0.71}$   | $\mathbf{4.02}$  |
> | $10^2$    | 194.84| 0.41      | 0.58    | 0.69  | 3.09  |
> | $10^3$     | 188.64| 0.52      | $\mathbf{0.33}$    | 0.65  | 2.22  |
>
> #### Table 3: Ablation Study on the Style Loss Weight (Dreambooth Finetuning on SD15)
> | $\lambda$  | FID   | Precision | Recall  | LPIPS  | CMMD   |
> |--------------|-------|-----------|---------|--------|--------|
> | 1 | 184.32| 0.29| 0.83   | 0.68  | 2.37|
> | 10| $\mathbf{227.20}$| $\mathbf{0.25}$| 0.72| $\mathbf{0.70}$  | $\mathbf{3.22}$   |
> | $10^2$| 194.32| 0.41| 0.50| 0.67  | 2.18  |
> | $10^3$| 180.21| 0.54| $\mathbf{0.41}$| 0.63| 1.69  |
>
> The ablation study results of the attack budget are shown in Table 4.  It is shown that as the budget increases, FID, LPIPS, and CMMD gradually increase, while Precision gradually decreases. When the budget is higher than 0.063, increasing the budget does not significantly improve the protection effect. This is because the difference in the generated images is already large enough. In addition, a too high budget will introduce visible protection noise, which will damage the quality of the protected image.
>
> #### Table 4: Ablation Study on the Attack Budget (Dreambooth Finetuning on SD15)
> | Budget | FID   | Precision | Recall  | LPIPS  | CMMD   |
> |-------|-------|-----------|---------|--------|--------|
> | 2/255| 165.76 | 0.39 | 0.52   | 0.60  | 1.09 |
> | 4/255 | 199.78 | 0.29 | 0.77  | 0.62  | 1.15 |
> | 8/255| 190.99 |  0.24 | $\textbf{0.50}$   | 0.63  | 2.61  |
> | 16/255| $\textbf{271.80}$ |$\textbf{0.04}$ |0.87| 0.67 |3.55 |
> | 32/255| 256.23   |  0.08 | 0.81   | $\textbf{0.70}$  | $\textbf{3.71}$  |
>
> For the ablation study of upscale loss weight $\eta$, $K_1$, and $K_2$, please refer to Response to Q1 of Reviewer 2.
>
> We also evaluate the influence of different purifiers used for the upscale loss. The result is shown in Table 5. We compute the upscale loss on three different settings (X2, X4, and X2+X4), and then use three different purifiers in the attack phase, including X2, X4, and a new purifier, SinSR [4]. The result shows that when both X2 and X4 are used for the upscale loss, the protection performance on the SinSR significantly improved. This shows that using an ensemble of purifiers can improve the protection strength.
>
> Table 5: Evaluate the Different Purifiers for the Upscale Loss
> | | X2  |  || X4  |  |  | X4+X2     |  |  |
> |-------|--------|---------|-----------|-----------|-----------|-------|-----------|-----------|-----------|
> |  | **FID** | **CMMD**| **PRE**   | **FID**   | **CMMD**  | **PRE**   | **FID**   | **CMMD**  | **PRE**   |
> | **X2**    | 197.637   | 2.9828    | 0.208     | 186.303   | 0.56  | 0.341 | 204.979   | 2.879     | 0.166     |
> | **X4**    | 188.308   | 2.205     | 0.25 | 191.35    | 2.748     | 0.2916 | 253.939   | 2.735     | 0.166 |
> | **SinSR** | 167.489   | 0.64  | 0.458 | 183.119   | 1.116  | 0.208  | 183.643   | 1.887 | 0.163 |
>
> ### Response to Q3
> Thanks for your suggestion. To defend against transformations such as crop and rotation, we utilize Expectation over Transformation (EoT) in our method (L194-195 in our paper). We have evaluated the crop and resize transformation in our paper (Table 1). To further evaluate the robustness of our method to transformation combinations, including crop and resize, rotation, and JPEG compression, we generate perturbations using SD 1.4 and then fine-tune the SD 1.5 model on the protected images. We compare our methods with SimAC. The training epochs for both methods are set to 20. Table 6 shows the results of the different combinations of transformations. The results show that we consistently outperform the SoTA baseline on different transformations, indicating that our method can effectively resist the combination of different transformations.
>
> Table 6: Evaluation on Diverse Transforms
> | Method  | Crop| | | Crop+Rotate  |  |  | Crop+JPEG Compress |  |     | Crop+Rotate+JPEG Compress |  | |
> |---------|---------|----------|--------|------------------|------------------|--------|----------|-----------|--------|-----------|-------|--------|
> |  | **FID** | **PRECISION**    | **CMMD** | **FID**  | **PRECISION**    | **CMMD** | **FID** | **PRECISION**    | **CMMD** | **FID**   | **PRECISION** | **CMMD** |
> | **SIMAC** | 271.463 |   0.04 | 2.146  | 202.784 | 0.167  | **2.908** | 190.32| 0.083| 3.583| 189.894 | 0.25  | 1.614|
> | **Ours** | **316.803**  | **0.00**    | **2.189** | **254.283** | **0.083**   | 2.379  | **321.339**| **0.00**| **3.971** | **245.415** | **0.00** | **3.636**  |
>
> Reference
> [1]  The unreasonable effectiveness of deep features as a perceptual metric. CVPR, 2018.
> [2]  Rethinking FID: Towards a Better Evaluation Metric for Image Generation. Arkiv2401.09603, 2024
> [3]  Improved Precision and Recall Metric for Assessing Generative Models. NIPS, 2019
> [4] SinSR: diffusion-based image super-resolution in a single step, CVPR2024

---

> ### Author Response · Authors · 2025-08-04
>
> Dear Reviewer YoSP,
>
> We hope this message finds you well.
>
> As the rebuttal period deadline is approaching, we would greatly appreciate it if you could kindly respond to our rebuttal. We are looking forward to receiving any further questions or suggestions for improvement. We thank you sincerely for reading our paper carefully.
>
> Best regards,
>
> Authors of NeurIPS 13927

---

> ### Author Response · Authors · 2025-08-08
>
> Dear Reviewer YoSP,
>
> I hope this message finds you well! I would like to take this opportunity to thank you once again for your thorough review and valuable comments on our paper. We have diligently addressed all of your concerns in our rebuttal submission.
>
> Considering the rebuttal deadline is approaching, I would kindly like to inquire if you have had the time to review our rebuttal.
>
> Thank you for your support and understanding. I look forward to your response!
>
> Best wishes,
>
> Authors of Paper 13927

---

### Decision · Program_Chairs · 2025-09-17

**Decision:**

Accept (poster)

**Comment:**

The paper proposes a defense system that adds small perturbations to protect digital artwork from being copied by text-to-image models. The reviewers found the idea important and timely, with the style loss and upscale loss being novel contributions that address both transferability and purification resilience. The experiments were thorough and showed strong results across DreamBooth, Textual Inversion, and multiple diffusion model versions. Weaknesses raised included unclear target style selection, lack of ablation on hyperparams, missing evaluation on combined real-world transformations, limited discussion of adaptive attacks, and some readability/formatting issues. The authors responded with detailed ablations, computational cost analysis, perceptual quality metrics, user study results, and multiple adaptive attack experiments. These additions significantly strengthened the submission, showing that the method is robust under varied conditions.

During discussion, reviewers initially leaned borderline, citing limited evaluation scope and lack of adaptive threat modeling. After rebuttal, most reviewers agreed the authors had addressed these concerns with extensive new experiments. One reviewer emphasized that adaptive attacks still deserve deeper treatment, but acknowledged the added experiments improved the situation. All reviewers converged on a borderline-accept stance, with updated justifications noting that while some limitations remain, the practical importance and clear novelty of styleguard make it a valuable contribution. Weighing the strong problem relevance, convincing empirical evidence, and the improved rigor after rebuttal, I recommend acceptance.